# Empowering Convolutional Neural Networks with MetaSin Activation

**Farnood Salehi**[1]  **Tunç Ozan Aydın**[1,3]  **André Gaillard**[3]

**Guglielmo Camporese**[1,2]*  **Yuxuan Wang**[3]

[1]Disney Research | Studios  [2]University of Padova  [3]ETH Zürich

## Abstract

RELU networks have remained the default choice for models in the area of image prediction despite their well-established spectral bias towards learning low frequencies faster, and consequently their difficulty of reproducing high frequency visual details. As an alternative, sin networks showed promising results in learning implicit representations of visual data. However training these networks in practically relevant settings proved to be difficult, requiring careful initialization, dealing with issues due to inconsistent gradients, and a degeneracy in local minima. In this work, we instead propose replacing a baseline network's existing activations with a novel ensemble function with trainable parameters. The proposed METASIN activation can be trained reliably without requiring intricate initialization schemes, and results in consistently lower test loss compared to alternatives. We demonstrate our method in the areas of Monte-Carlo denoising and image resampling where we set new state-of-the-art through a knowledge distillation based training procedure. We present ablations on hyper-parameter settings, comparisons with alternative activation function formulations, and discuss the use of our method in other domains, such as image classification.

## 1   Introduction

Deep convolutional neural networks are highly proficient in a wide array of tasks including image prediction applications. While there is no scarcity of inventive methods when it comes to designing convolutional blocks (e.g. [25]), another crucial design element, namely the activation function, has seen little change over the years. The RELU activation, despite (and arguably because of) its simplicity has remained the default choice for majority of the models in the area of image prediction.

It is often desirable for models that produce visual results to produce sharp imagery with abundant high frequency details. The widespread use of RELU networks for image prediction applications is somewhat surprising given their spectral bias towards learning low frequencies first [33], and consequently their difficulty in reproducing high frequency visual details. While sin activations have been shown to be superior in simple overfitting experiments on image reconstruction [39], empirical evidence showed that training sin networks in practically relevant settings is challenging even when employing meticulous initialization schemes. While some exploration on using sin activations with convolutional networks exists [8], their use has been mostly limited to fully connected architectures.

In this work we make the following contributions:

---

*This work was done while the author was an intern at Disney Research | Studios

37th Conference on Neural Information Processing Systems (NeurIPS 2023).

- We shed light on challenges associated with training networks with $\sin$-based activations through an extensive set of experiments, and address those by formulating the METASIN activation coupled with a stable training procedure.
- We show that METASIN networks consistently enable improvements over alternatives when tested on comparable architectures in a variety of image prediction applications, including Monte-Carlo denoising and image resampling where we set new state-of-the-art.
- We present a wide range of ablations on hyperparamter settings, comparisons with various alternative activations, and discuss the use of METASIN networks in other domains such as classification and neural signal representations.

## 2   Related Work

**Activation Functions** In addition to standard non-polynomial functions such as Sigmoid, $\tanh$, $\sin$, RELU, and $\exp$, numerous alternative activation functions have been explored in the literature. Previous work explored RELU variants such as Leaky RELU [26], Parametric rectified linear unit (PReLU) [14], and Gaussian error linear unit (GELU) [17]. Other authors presented activations with exponential components, such as the Exponential linear unit (ELU) [11], Scaled exponential linear unit (SELU) [22], as well as Sigmoid variants such as Softplus [12], and SWISH [34]. A comparative study of these activations re-affirmed RELU as a strong baseline, while suggesting SWISH might be a better alternative for image classification [34]. Another study [25] reported no accuracy change when switching from RELU to GELU in image classification. The aforementioned body of work has mostly focused on classification experiments, whereas their use in image prediction models was left mostly unexplored.

Previous investigations [34] pointed out periodic activations as a promising direction, which was later explored in the form of $\sin$ activations for implicit neural representations of continuous signals [39]. More recent work proposed modulating the amplitudes of $\sin$ functions using a secondary fully connected network [27]. Another $\sin$ variant is Snake [49], which takes the form $x + sin^2(x)$ and focuses on improving extrapolation of periodic functions. Finally, several papers explored combining multiple primitive functions, such as Mish ($x \tanh (\text{softplus}(x))$ [30], SinLU ($\sin$ and Sigmoid) [32], and activation ensembles [13]. Preliminary experiments [39] suggest that activations alternative to RELU, with the exception of $\sin$, tend to perform sub-par on image reconstruction. The popularity of RELU networks in image prediction tasks contradicts the theoretical analysis presented in previous work [33] that established the spectral bias of RELU networks towards learning low frequencies faster. Later work [7] re-affirmed this finding through a neural tangent kernel [20] analysis.

**Image Resampling** Tasks such as correcting lens distortion, retargeting, and upsampling are some image resampling examples that are considered as key operations in real-world computer vision and image processing pipelines. Recent neural methods [5, 40, 9] have been shown to outperform traditional techniques, such as upsampling with a fixed kernel. These neural networks often comprise convolutional blocks [16, 19] with RELU activations. Typically utilizing deep networks, current resampling methods can learn complex representations of input images and produce high-quality resampled outputs. In this work, we employ METASIN to improve upon a recent state-of-the-art resampler [5].

**Monte-Carlo Denoising** Generating high quality rendered images may require prohibitive amount of compute resources. This can be alleviated by stopping the rendering process prematurely and employing a specialized denoiser to remove the residual noise. Neural denoisers have demonstrated superior performance compared to classical denoising techniques [4, 44, 48]. These denoisers often leverage U-Net with RELU activation [37], which allows them to process noisy images across multiple scales. We likewise demonstrate the use of METASIN in such U-Nets and achieve significant improvements in denoising quality.

## 3   METASIN Activation

Let us define a generic neural network $g(\mathbf{x})$ of depth $L$ that predicts a target image $\mathbf{y} \in \mathbb{R}^d$ given input $\mathbf{x} \in \mathbb{R}^{d'}$ as:

$$\mathbf{y} \approx g(\mathbf{x}) = \left( \phi^{[L]} \circ T^{[L]} \circ \phi^{[L-1]} \circ T^{[L-1]} \circ \ldots \circ \phi^{[1]} \circ T^{[1]} \right)(\mathbf{x}), \qquad (1)$$

where $T^{[l]}$ denotes a linear transformation that results from either a convolutional or fully connected layer. We would like to find a set of activations $\phi^{[1]} \ldots \phi^{[L]}$ such that they minimize the average reconstruction error $\sum_i^n \mathcal{L}\left(g\left(\mathbf{x}^{(i)}\right) - \mathbf{y}^{(i)}\right)$ over a dataset $\{\mathbf{x}^{(i)}, \mathbf{y}^{(i)} \in [1, n]\}$.

In hypothetical limit cases where the network $g$ has infinite width or depth, few constraints on activation functions are necessary for the resulting architecture to have the universal approximation capability [18]. However, in practice the selection of the activation functions can play a significant role in the training of neural networks, specifically in the resulting model's prediction accuracy and, in case of image prediction models, the visual characteristics of the predictions. While it is unclear how a principled method can be developed for rank ordering all possible selections of activations, empirical evidence suggests that multi-layer perceptron (MLP) networks with sin activations exhibit desirable characteristics both in expressiveness and their ability to reproduce rich high frequency details [39], especially in comparison to RELU networks that suffer from spectral bias towards reproducing low frequencies [33].

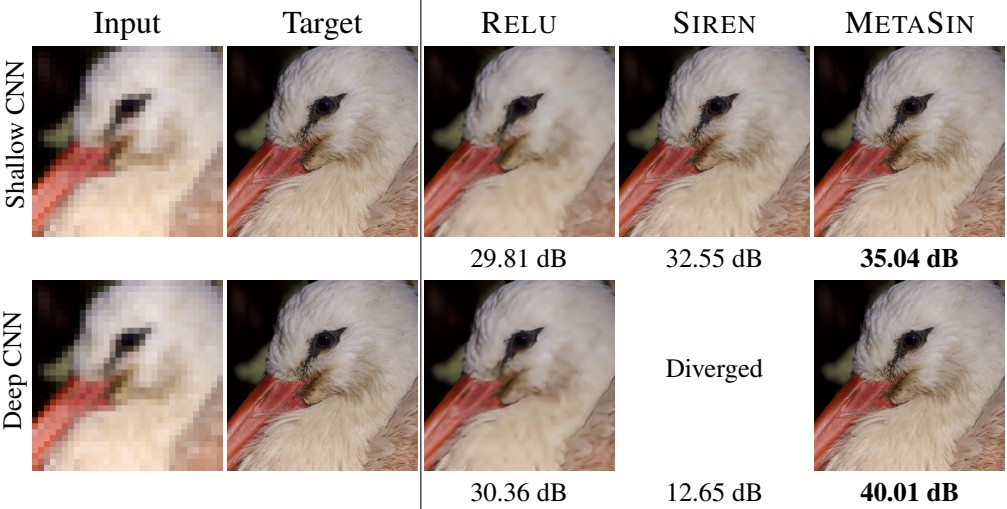

Figure 1: Peak Signal-to-Noise Ratio (PSNR) comparison of different activations employed on CNNs to overfit an image from its downsampled version (with downsampling factor $16\times$). Notably, while the shallow sin network performs better than its RELU counterpart, the deep sin network fails to converge, demonstrating the fragility of training deep convolutional networks with sin activations. See Appendix B for details.

However, practical experience also revealed the difficulty of training deep convolutional sin networks in real-world scenarios. Figure 1 shows an illustrative example that on one hand demonstrates the clear advantage of sin activations over RELU when training shallow networks, but on the other hand illustrates the fragility of training deeper sin networks that are closer representatives of real-world models. To examine further we express an intermediate segment from a sin activated network as:

$$\mathbf{a}^{[l]} = \sin\left(T^{[l]}\left(\mathbf{a}^{[l-1]}\right)\right), \tag{2}$$

where $\mathbf{a}^{[l]}$ is the output of layer $l$ after applying the corresponding linear transformation and activation function. We can also express the usual parameterization of sine waves as $f(\mathbf{z}; c, f, p) = c\sin(f\mathbf{z} + p)$, where $c$, $f$, and $p$ are the amplitude, frequency, and phase of the sine wave that we collectively refer as the *shape parameters*.

In overparameterized networks the initial parameter values tend to change very slowly [10] (also known as lazy training). Therefore ensuring the selection of plausible initial values for the shape parameters in sin networks is crucial for training. Some exploration has been done in this direction: For instance, SIREN [39] introduces a scaling constant $w_0$ that is applied to the weight matrix $W$ of the first layer of the underlying fully connected network, i.e. $\sin(w_0 W\mathbf{x} + \mathbf{b})$, whereas recent work explored introducing an explicit amplitude parameter whose value is predicted through a secondary neural network [27]. Yet initialization remains a challenge as in practice only a very narrow range of values leads to plausible average loss (Refer to Figure 10 in Appendix C).

In this work, we instead introduce explicit shape parameters for directly controlling amplitude, frequency, and phase. These parameters are completely disentangled from other network parameters, and can be initialized intuitively by reasoning about the shape of resulting sine wave. Moreover, in order to provide better initial coverage of plausible ranges of the shape parameters, we construct a composite periodic function by linearly combining $K$ sine waves with individual shape parameters:

$$\sum_{j=1}^{K} c_j^{[l]} \sin\left(\mathbf{z}^{[l]} f^{[l]} + p^{[l]}\right), \text{ where } \mathbf{z}^{[l]} = T^{[l]}\left(\mathbf{a}^{[l]}\right). \tag{3}$$

The shape parameters are shared among individual channels of layers in convolutional neural networks, and are optimized through backpropagation along with other network parameters[2]. In our experiments we observed that initializing the frequency parameters $f_j^{[l]}$ of the $K$ constituent sine waves of the composite function to cover a large range of frequencies consistently yields improved results, without necessitating lengthy trial and error procedures that are often not feasible when training large models in practice (Refer to Appendix C for an illustrative example).

Beyond the initialization of shape parameters, the remaining known causes of the difficulty when training $\sin$ networks are inconsistent gradients due to the complex shapes that activations can take, and the large degeneracy in local minima caused by symmetries [49]. We empirically found that including an additional RELU component to the composite sine wave function discussed before stabilizes training. With this modification our proposed activation, which we call METASIN, can be expressed as follows:

$$\phi(\mathbf{z}^{[l]}) = c_0^{[l]}\text{RELU}\left(\mathbf{z}^{[l]}\right) + \sum_{j=1}^{K} c_j^{[l]} \sin\left(f_j^{[l]}\mathbf{z}^{[l]} + p_j^{[l]}\right). \tag{4}$$

Equation 4 can efficiently be implemented as a module in common deep learning frameworks, which then can be used to replace activation functions of an existing state-of-the-art network architecture. The trainable parameters $\theta = \{c_0^{[l]}, c_j^{[l]}, f_j^{[l]}, p_j^{[l]}; j \in [1,K], l \in [1,L]\}$ are optimized through backpropagation without requiring an additional network for predicting these particular parameters, or any other changes to the underlying architecture. This non-intrusive property enables a simple three-step procedure, where (i) we identify a state-of-the-art image prediction model, (ii) replace its existing activation functions with METASIN, and (iii) re-train the resulting METASIN network using a knowledge distillation scheme we discuss in the following section.

## 3.1 Training

In order to train convolutional METASIN networks, we initialize the shape parameters as $c_0 = 1$, $c_j^{[l]} = 0$, $f_j^{[l]} = j$, and $p_j^{[l]} = \mathcal{U}(0,\pi)$, for $j \in [1,K]$, and $l \in [1,L]$ by default. This initialization on one hand forces METASIN to initially assume the shape of RELU, and gradually introduce $\sin$ components during the course of the training, while on the other hand covering a broad range of initial frequency and phase values (See Appendix G). Using the default initialization we were able to reliably avoid typical issues encountered while training $sin$ networks.

It is often easier to train a shallower network compared to a deeper one as supported by previous studies [16]. This observation also applies to $\sin$ activations, as demonstrated in Figure 1, where a shallow SIREN network can be effectively trained while a deep SIREN network encounters difficulties. To address this, we employ feature knowledge distillation (KD) from a teacher network [35] to provide auxiliary signals during the training of the student METASIN network. We first train a RELU model as the teacher (or, as it often happens in practice, use an already existing baseline RELU model) and use its intermediate feature maps as supervisory signals. This approach allows us to train the shallower blocks of the student METASIN network individually, instead of propagating gradients throughout the entire network. We refer to this phase of training as *KD-Bootstrapping*, which comprises approximately $5-10\%$ of the total training iterations for the METASIN network. Following the initial KD-Bootstrapping phase, the student METASIN network is trained using the same configuration as the baseline network. This approach enhances the learning process and helps improve the performance of the student network.

---

[2]With a certain selection of parameters the resulting composite function can be shown to be equivalent to a Fourier Series, and hence the hyperparameter $K$ can be thought as controlling the expressiveness (Appendix D).

## 3.2 Efficient Implementation

While equation 4 is straightforward to implement in both PyTorch and Tensorflow using the corresponding Python APIs, such native implementations end up being inefficient compared to executing a RELU function. For performance critical applications, efficiency issues can be alleviated through writing a customized operation in CUDA (Appendix F). In Table 1 we compare the latency induced by the native PyTorch implementation against our customized METASIN operator with fused CUDA kernel functions. Our optimized implementation significantly reduces the overhead of the native implementation. During forward pass, executing a METASIN with $K = 10$ components requires less than three times the time it takes to execute a RELU function, whereas the backward computation is only slightly more expensive than a RELU activation. Our efficient implementation also uses roughly the same amount of memory during execution as a RELU activation, making METASIN networks feasible to use in practice on the same hardware that a comparable RELU network is designed for.

| Module | Impl. | Forward (rel) | Backward (rel) |
|---|---|---|---|
| METASIN | PyTorch Native | 26.1x | 1.58x |
| | PyTorch CUDA | **2.9x** | **1.12x** |

Table 1: Latency of executing a METASIN activation with $K = 10$ using native vs. efficient implementations, relative to the latency of executing a single RELU activation on Nvidia RTX 3090.

# 4 Experiments

We present experiments where we apply the three step procedure discussed at the end of Section 3. Specifically, we switch an existing architecture's activations to METASIN, and re-train the resulting new model from scratch using KD-Bootstrapping.

| | Activation | | | | | | | | | |
|---|---|---|---|---|---|---|---|---|---|---|
| Factor | RELU | RELU$_E$ | SNAKE | MISH | SIREN | SIREN KD-B | MRELU | MRELU KD-B | METASIN | METASIN KD-B |
| $P_{\times 2}$ | 33.03 | 33.04 | 32.95 | 32.96 | 31.75 | 32.76 | 32.78 | 33.02 | 33.02 | **33.26** |
| $P_{\times 3}$ | 29.36 | 29.36 | 29.16 | 29.06 | 28.18 | 28.98 | 29.16 | 29.36 | 29.35 | **29.58** |
| $P_{\times 4}$ | 27.09 | 27.10 | 26.97 | 26.89 | 26.20 | 26.83 | 26.97 | 27.10 | 27.11 | **27.29** |

Table 2: Comparison of PSNRs obtained from various image resampling models. See text for details.

## 4.1 Image Resampling

In image resampling our approach builds upon the state-of-the-art network proposed by Bernasconi et al. [5], which incorporates ProSR [45] feature extractor with 3 residual blocks, a resampling layer, and a prediction layer. The network takes an input image and a warp grid and performs image resampling. We trained baseline networks following the exact same training procedures and dataset as described in [5]. The METASIN network is obtained by replacing RELU activations of the feature extractor with METASIN with $K = 10$ and $f_j^{[l]} = j$ at initialization. All other parameters are initialized as described in Section 3.1. The transition to METASIN activations increases the amount of compute by 3% (and similarly the wall-clock inference time by 3%) compared to the baseline RELU network. To rule out any improvement due to this increase in computation, we accordingly increase the number of channels of baseline RELU network, which we label as RELU$_E$.

We evaluate the models on three predetermined sets of projective transforms with average local scaling factors of 2 ($P_{\times 2}$), 3 ($P_{\times 3}$) and 4 ($P_{\times 4}$). We report the PSNR scores of the resampled images in Table 2, as well as visual examples in Figure 2 We use KD-Bootsrapping during the first 10% of the training procedure. On average the METASIN network with KD-Bootstrapping improves over the RELU$_E$ baseline by 0.2dB, setting a new state-of-the-art in image resampling. We also conducted KD-Bootstrapping on RELU, MISH and SNAKE networks, but observed no improvement.

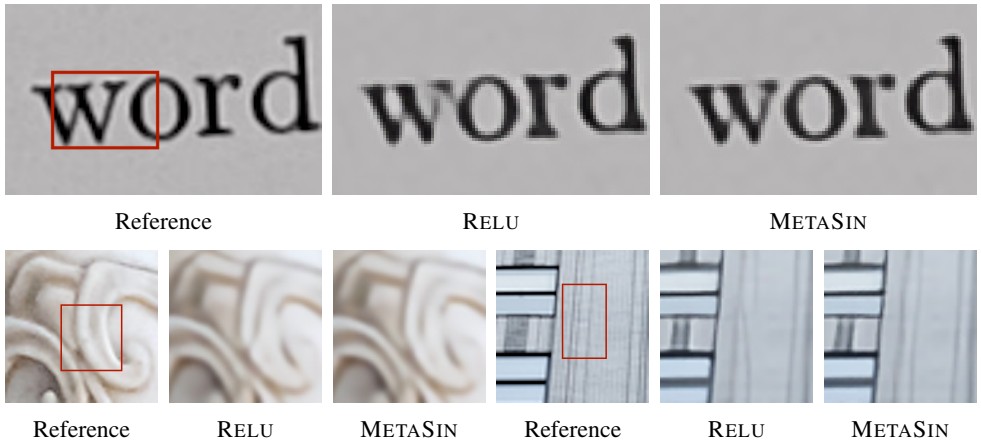

Figure 2: Example upsampling results comparing the state-of-the-art model with RELU activations [5] and the corresponding METASIN network.

This outcome suggests that the training process for these networks is inherently stable, making KD-Bootstrapping unnecessary for these activations. Refer to Appendix K for additional results.

## 4.2 Denoising Monte-Carlo Rendered Images

Next, we present our experiments in denoising Monte-Carlo rendered images. Our base denoiser network is a U-Net architecture used in [38] and we use the same procedure to generate noisy and reference renderings. Since we focus on direct image prediction models rather than kernel denoising as in [38], our U-Net denoiser (DPCN) directly predicts pixel values of the clean image (For completeness, the results of kernel denoising (KPCN) are presented in Appendix M). During evaluation, we denoise images rendered with $[2, 4, 8, 16]$ samples per pixel and use SMAPE, FLIP [2], $1-\text{MS\_SSIM}$, and $1-\text{SSIM}$ as our metrics. Our best METASIN network is configured with $K = 5$ and the frequencies are initialized as $f_j^{[l]} = j/2$. We use KD-bootsrapping during the first 5% of the training procedure. We present a summary of various METASIN network configurations in Table 3, and selected visual examples in Figure 3.

| Metric | DPCN | METASIN-5 $(f_j = j)$ | METASIN-10 KD-B (Glorot) | METASIN-5 KD-B $(f_j = j)$ | METASIN-5 KD-B $(f_j = j/2)$ |
|---|---|---|---|---|---|
| SMAPE | 3.351 | 3.16 (-5.7%) | 3.118 (-6.95%) | 3.088 (-7.85%) | **3.081 (-8.06%)** |
| FLIP | 1.084 | 0.991 (-8.58%) | 0.972 (-10.33%) | 0.976 (-9.96%) | **0.965 (-10.98%)** |
| 1-MS_SSIM | 3.106 | 2.838 (-8.63%) | 2.784 (-10.37%) | 2.757 (-11.24%) | **2.733 (-12.01%)** |
| 1-SSIM | 6.895 | 6.385 (-7.4%) | 6.227 (-9.69%) | 6.146 (-10.86%) | **6.112 (-11.36%)** |

Table 3: Results of various METASIN configurations. All relative values are in relation to the DPCN baseline. The set of test samples contains all samples with spp $\leq 16$. The frequency initializations are given in parenthesis, which shows that a slight improvement can be made by initializing as $f_j = j/2$ rather than the default $f_j = j$. Switching to Glorot initialization results in reduced accuracy despite the doubled $K$. Introducing KD-Bootstrapping during training significantly improves accuracy.

## 5 Discussion

In this section we present ablation experiments and an exploratory study on using METASIN activations for image classification and implicit image representation using MLPs. We present preliminary investigations of further applications of METASIN activation in Appendix A

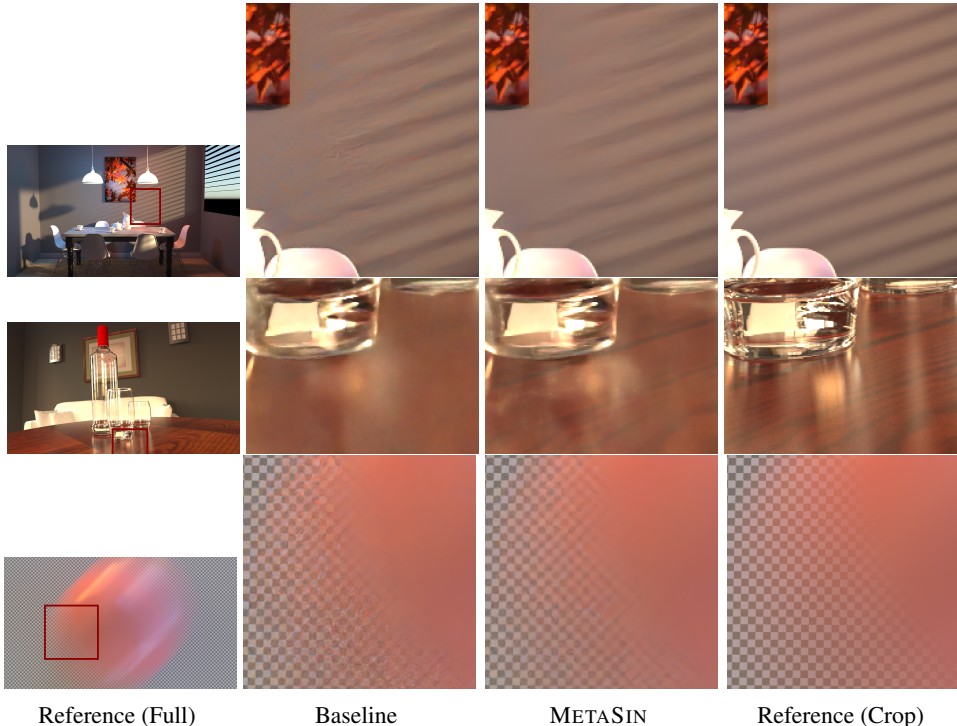

| Reference (Full) | Baseline | METASIN | Reference (Crop) |

Figure 3: Monte-Carlo denoising results. The first row shows results produced with an input noisy images rendered at 4 samples-per-pixel, whereas the input in the remaining two rows are rendered at 16 samples-per-pixel.

## 5.1 Comparison with Alternative Activations

| Metric | DPCN | MRELU-10 | MRELU-10 KD-B | MISH | METASIN-10 | METASIN-10 KD-B |
|---|---|---|---|---|---|---|
| SMAPE | 3.351 | 3.472 (+3.61%) | 3.254 (-2.89%) | 3.293 (-1.73%) | 3.346 (-0.15%) | **3.118 (-6.95%)** |
| FLIP | 1.084 | 1.096 (+1.11%) | 1.031 (-4.89%) | 1.054 (-2.77%) | 1.06 (-2.21%) | **0.972 (-10.33%)** |
| 1-MS_SSIM | 3.106 | 3.397 (+9.37%) | 2.963 (-4.6%) | 3.011 (-3.06%) | 3.066 (-1.29%) | **2.784 (-10.37%)** |
| 1-SSIM | 6.895 | 7.044 (+2.16%) | 6.589 (-4.44%) | 6.896 (+0.01%) | 6.736 (-2.31%) | **6.227 (-9.69%)** |

Table 4: Comparison of various alternative activations on the DPCN model discussed in Section 4.2. All parameters are initialized using Glorot initialization. See text for discussion.

We ran several experiments to assess the comparative improvements that can be achieved by using alternative activations instead of METASIN. In addition to RELU, we also tested SNAKE ($x + \sin^2(x)$) in [49]), MISH ($x \tanh(\ln(1 + e^x))$ in [30]), and MRELU in which we linearly combine several RELU activations in the form $\sum_{j=1}^{K} c_j \text{RELU}(x + b_j)$ (Similar to Adaptive Piecewise Linear Units [3]). The SNAKE activation [49] performs slightly worse than the RELU baseline in the image resampling application, which in turn falls below the corresponding METASIN activated network (Table 2 - $3^{nd}$ col). In case of Monte-Carlo denoising it was not possible to train the U-Net model with SNAKE activations as the training diverged. MISH similarly performs slightly worse than both RELU and SNAKE in image resampling (Table 2 - $4^{th}$ col).

In Monte Carlo denoising the MISH activated network performs better than the RELU baseline by 3% in terms of MS_SSIM, however still falls significantly behind the METASIN activated network that achieves 10% improvement in the same experiment (Table 4).

The MRELU activation with $K = 10$ performs significantly worse than the RELU baseline both in denoising and resampling task (Table 4 - $3^{rd}$ col and Table 2 - $7^{th}$ col). We also tested applying

KD-Bootstrapping to MRELU which made a big improvement in all metrics, still, however, scoring far behind the corresponding METASIN network.

Overall, METASIN activations ended up consistently yielding notably better results than the activations we tested in all our experiments. In Appendix H we present further comparisons with ensemble activations.

## 5.2 The Effect of Hyperparameters

The METASIN activation as formulated in Equation 4 has a single hyperparameter $K$ that determines the number of constituent sine waves. While in our experiments we often set $K = 10$, we also run ablation experiments to measure the effect of different $K$ values. In these experiments we used the exact same settings as in Section 4.2, except varying the $K$ parameter. In Table 5 we show the effect of changing $K$ to 1, 5, and 12, compared to the baseline where $K = 10$. Not surprisingly, lowering $K$ results in consistently worse results according to all four metrics we tested. These findings also show that the results that we presented in Section 4.2 can be further improved simply by setting $K = 12$ (See Appendix I for further discussion). Aside from $K$, another decision to be made when using METASIN activations is whether to train with KD-Bootstrapping or not. While in toy examples we haven't noticed any difference between the two alternatives, the last two columns in Table 4 show a comparison with and without KD-Bootsrapping, which clearly demonstrates the benefit of employing KD-Bootsrapping in a real-world scenario.

| Metric | METASIN-10 KD-B | METASIN-1 KD-B | METASIN-5 KD-B | METASIN-12 KD-B |
|---|---|---|---|---|
| FLIP | 0.972 | 0.997 (+2.57%) | 0.979 (+0.72%) | **0.967 (-0.51%)** |
| 1-MS_SSIM | 2.784 | 2.825 (+1.47%) | 2.805 (+0.75%) | **2.767 (-0.61%)** |

Table 5: Comparison of DPCN networks (discussed in Section 4.2) with various $K$ parameters trained with KD-Bootstrapping and Glorot frequency initialization.

## 5.3 Image Classification using Convolutional METASIN Networks

We ran several preliminary experiments for exploring the use of METASIN networks outside our target domain of image prediction tasks. Here we report the results of various experiments in a classification setting, where we trained various student Wide-ResNets using supervision from comparably larger Wide-ResNets that we used as teacher networks. As the teacher network in this case is a larger network with higher accuracy, replicating its predictions becomes a reasonable objective. Therefore, we employ feature knowledge distillation throughout the entire training procedure. The Wide-ResNet architecture [47] follows the same architectural design as ResNet [15], except that the number of channels in each residual block is expanded.

The METASIN activated students are identical to their RELU activated counterparts, except the activations are replaced by METASIN with $K = 8$. We ran our experiments on the CIFAR100 [23] dataset, which consists of 50K training and 10K validation images with resolution $32 \times 32$. The student networks were trained for 300 epochs using standard cross-entropy loss following the knowledge distillation methodology discussed in [36]. While not setting a new state-of-the-art, Table 6 shows that the METASIN student networks consistently outperform their RELU counterparts, and in some cases also the corresponding SWISH student networks, motivating further investigation of using METASIN activations in classification tasks. However, we note that the accuracy of METASIN models are less competitive compared to alternatives when models are trained from scratch without knowledge distillation (Appendix J).

## 5.4 METASIN with Multi-Layer Perceptrons

While in this work we focused on applications of METASIN with convolutional neural networks, we also ran several exploratory experiments where we used METASIN activations with MLPs to perform implicit neural representation tasks as in [39]. We refer the reader to Appendix A.1 for a discussion on video overfitting, where the METASIN network achieves 1-4 dB PSNR improvement over the SIREN

| | RELU | METASIN | SWISH | RELU | METASIN | SWISH | RELU | METASIN | SWISH | RELU | METASIN | SWISH |
|---|---|---|---|---|---|---|---|---|---|---|---|---|
| Teacher | WRN-40-2 | | | WRN-40-4 | | | WRN-28-2 | | | WRN-40-2 | | |
| Student | WRN-40-1 | | | WRN-40-1 | | | WRN-16-2 | | | WRN-16-2 | | |
| Teacher Acc. | 76.08 | | | 78.58 | | | 74.82 | | | 76.08 | | |
| Activ. | RELU | METASIN | SWISH | RELU | METASIN | SWISH | RELU | METASIN | SWISH | RELU | METASIN | SWISH |
| Val. Acc. | 73.39 | **73.74** | 73.28 | 70.53 | **72.55** | 72.54 | 71.76 | 72.33 | **72.98** | 73.65 | **74.10** | 72.60 |

Table 6: Comparing RELU student networks with METASIN activated networks with otherwise identical architecture, both trained through knowledge distillation from the same RELU activated teacher.

baseline with noticeably less visual artifacts. In Appendix A.3 we discuss a voxel grid overfitting experiment where the METASIN network exhibits significantly improved detail reconstruction over the RELU baseline. We also present a preliminary novel view synthesis experiment where we compare a baseline Neural Radiance Field (NeRF) method [29] with a corresponding METASIN model with $K = 10$.

We present an example result from this experiment in Figure 4, and refer the reader to Appendix A.2 for more details. Despite not utilizing positional encoding, the METASIN network surpasses vanilla NeRF in performance. It has been observed that Fourier features[42], which bear similarities to positional encoding, aid in learning high-frequency functions. This experiment implies that such feature engineering may not be crucial, as our METASIN network achieves comparable or even superior performance. While this result is intriguing, we acknowledge recent techniques that leverage latent codes in the form of memory, combined with specialized data structures [31, 41], which can significantly enhance vanilla NeRF in terms of performance and training time. While not conclusive, we hope the initial results we share in this section motivate further investigation of METASIN activations in the context of other architectures and application areas.

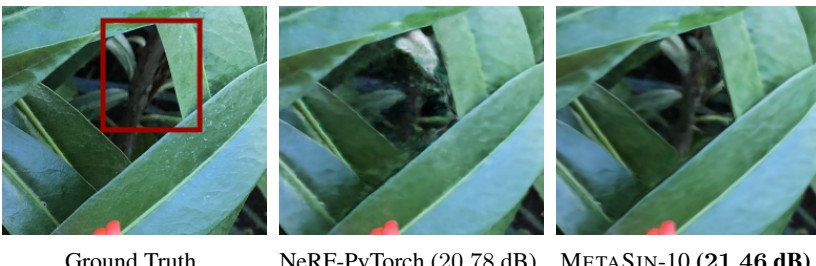

Ground Truth      NeRF-PyTorch (20.78 dB)      METASIN-10 **(21.46 dB)**

Figure 4: Zoomed qualitative results on scene *flower* [28]. Comparison between ground-truth, NeRF [29] and METASIN-10 model, with corresponding PSNR scores in parenthesis.

## 6    Conclusion and Limitations

We presented a novel activation function with trainable parameters that improves upon state-of-the-art convolutional image prediction models. Our work has several limitations. While METASIN networks can reliably be trained without it, training with KD-Bootstrapping helps the network to eventually converge to better minima. While it is not uncommon when training a METASIN network to have access to a comparable trained RELU network, nevertheless this requirement might lead to some extra effort when a pre-trained base model is not available. Additionally, in our current formulation (Equation 4) the number of sin components $K$ is a hyperparameter that we set manually at design time. We performed preliminary experiments with determining $K$ automatically in an adaptive fashion, which would be an interesting direction to study more in the future.

The METASIN function is only $C^0$ in contrast to the infinitely differentiable SIREN. The $C^\infty$) property allows SIREN networks to be trained using loss functions on the gradient or Hessian of the network. This distinct characteristic of the SIREN network is used later in [24] to automatically integrate the function that appears in neural volume rendering. The lack of a gradient loss in METASIN networks seems to limit the performance of METASIN activations for applications such as implicit 3D shape representations (Appendix N). It would be interesting to tackle this inherent limitation of METASIN networks in future work.

## Broader Impact

Fundamentally, neural network blocks comprise a linear transformation and a non-linear activation. Our work introduces a new activation function that enables significant prediction accuracy improvements over previous alternatives. Being at such a fundamental level, our proposed contribution thus can be applied to a wide array of neural network types that perform a variety of tasks and facilitate improvements upon existing state-of-the-art results. The resulting improved models may be leveraged for both positive and negative ends.

## Acknowledgements

We thank Xianyao Zhang, Marios Pappas, Michael Bernasconi, and Aziz Djelouah for their help in Monte-Carlo denoising and resampling experiments, Katarina Tothova for proof-reading our manuscript. We also thank Wig42 for the "RoomChair" and "LivingRoom" images, Xianyao Zhang for the "furball" image in Figure 3, nacimus for "Can" image, SlykDrako for the "BedRoom" image in Figure 15. The dataset for the Nerf experiments were provided by Matthew Tancik and Ben Mildenhall.

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

# Appendix

## A    Use of METASIN Beyond Image Prediction

In addition to our main results presented in Section 4 of the paper, we also performed various exploratory experiments to investigate further application cases of the METASIN activations. The experiments cover image classification where we show favorable results of using convolutional METASIN networks over baseline RELU networks, as well as various overfitting experiments to explore the use of METASIN activations with MLPs. We present our preliminary findings here with the hope of motivating further investigation in these directions.

In all MLP experiments, we use METASIN with $K = 10$ sine components, and distribute the frequencies evenly across the range $[1, 35]$. The initialization of the remaining parameters follows the description provided in Section 3.

| Target | SIREN | METASIN |
|--------|-------|---------|

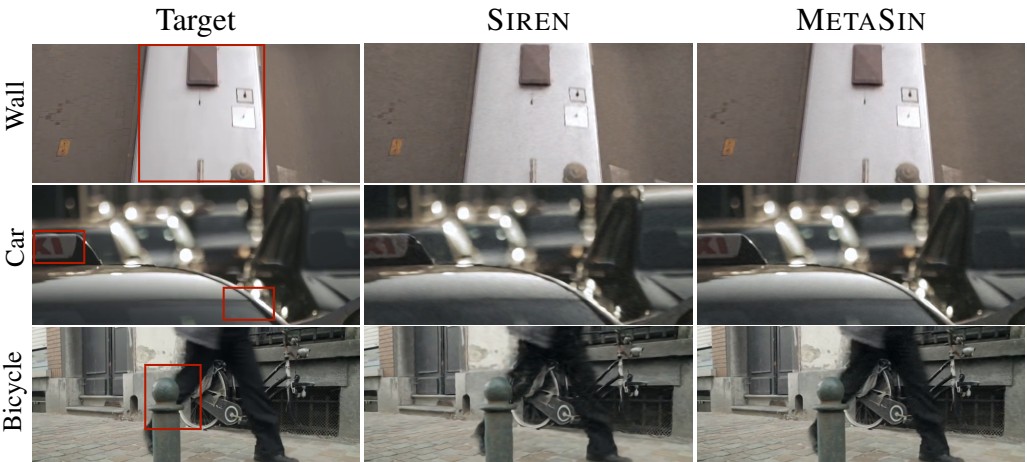

Figure 5: Visualization of selected reconstructed frames of the video. To fully appreciate the details and visual cues presented in the figure, we recommend visualizing the figures in color and zooming in for a more comprehensive analysis.

| Video Dataset | Activation | Model | Channels | Depth | MSE ↓ | PSNR (dB) ↑ |
|---------------|-----------|-------|----------|-------|-------|-------------|
| Cat | SIREN | MLP | 1024 | 3 | $2.60 \cdot 10^{-3}$ | 32.26 |
| | METASIN | MLP | 1024 | 3 | $\mathbf{1.96 \cdot 10^{-3}}$ | **33.32** |
| Bikes | SIREN | MLP | 1024 | 3 | $1.69 \cdot 10^{-3}$ | 34.50 |
| | METASIN | MLP | 1024 | 3 | $\mathbf{5.58 \cdot 10^{-4}}$ | **39.32** |

Table 7: Results on implicit video representation.

### A.1    Video Overfitting using METASIN MLP

In this section, we present the overfitting experiments on videos using an MLP network with 3 hidden layers and 1024 neurons to overfit a video. The input is a 3 dimensional vector consisting of a frame ID and pixel coordinate, and the output is the RGB value of the corresponding pixel. The METASIN network improves the PNSR by 1-4 dB compared to the SIREN baseline as shown in Table 7.

Overall the *Cat* dataset has more fine details, making it more challenging to accurately represent these intricate details. Consequently, the PSNR for the *Cat* dataset is comparatively lower in comparison to the *Bikes* dataset. The results are depicted in Figure 5. While RELU networks struggle to represent the signal, both SIREN network and METASIN can reconstruct the frames. Notably, the METASIN network reconstructs more details, see Figure 5 (Bicycle) and has less noise, as we see in Figure 5 (Wall and Car). It is worth mentioning that various techniques, such as positional encoding or Fourier

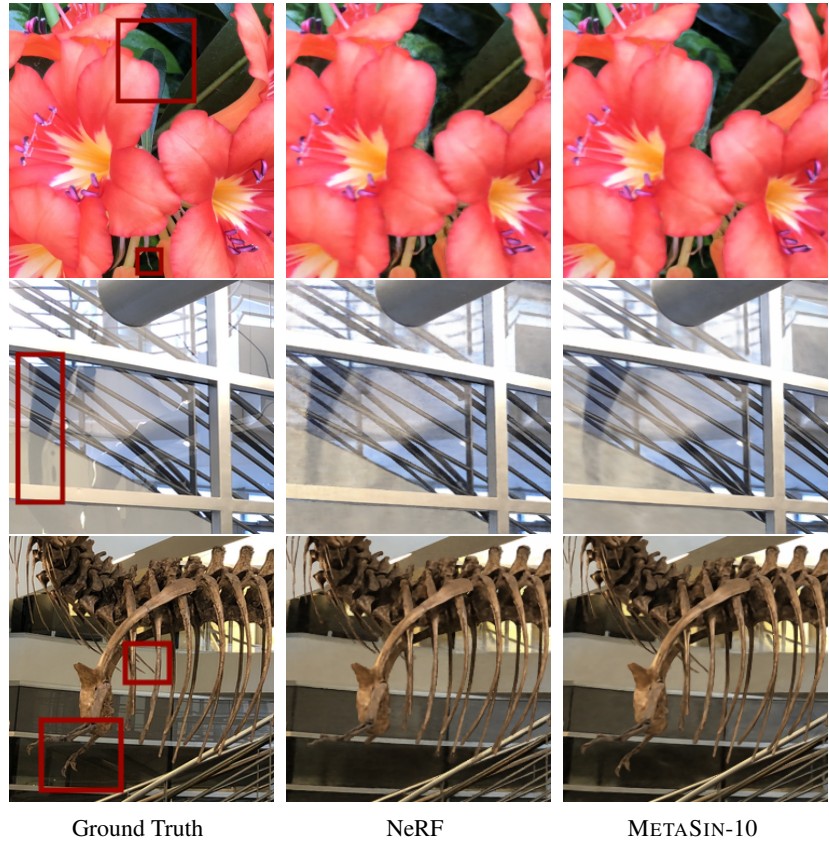

| Ground Truth | NeRF | METASIN-10 |

Figure 6: Zoomed qualitative results on static forward scenes [28]. Comparison between ground-truth, NeRF and METASIN-10 model.

features [43, 42], as well as utilizing latent codes [31], have shown significant improvements in RELU networks. However, exploring the impact of these techniques when applied in conjunction with METASIN activation is a direction we defer to future work.

| Method | Leaves | Orchids | Flower | T-Rex | Horns |
|---|---|---|---|---|---|
| NeRF [29] | 20.92 | 20.36 | 27.40 | 26.80 | 27.45 |
| METASIN-10 | **21.46** | **20.52** | **28.20** | **26.81** | **27.98** |

Table 8: Per-scene PSNR↑ comparison. In this table, we report the results obtained by running NeRF PyTorch implementation, based on [46] and previously used in [21, 6], and the results of our proposed METASIN. Our method achieves higher PSNR in all tested scenes.

### A.2 Novel View Synthesis using METASIN MLP

*Neural Radiance Fields* [29] (NeRF) enable synthesizing novel views with only a sparse set of input views. NeRF methods exploit *positional encoding* to map the input into high-frequency space, which is the key to high-quality scene rendering, which we drop in our experiments and replace the standard RELU with a 10 components METASIN-10 activation. Frequencies and phases are initialized uniformly over $[1, 35]$ and $[0, 2\pi]$. The experiment setting is the same as [29], except that we extend the training process to 250K iterations instead of 200K, in order to make sure that METASIN models converge well. Our implementation is based on [46], The results are reported in Table 8. We also provide a comparison of qualitative results in Figure 6.

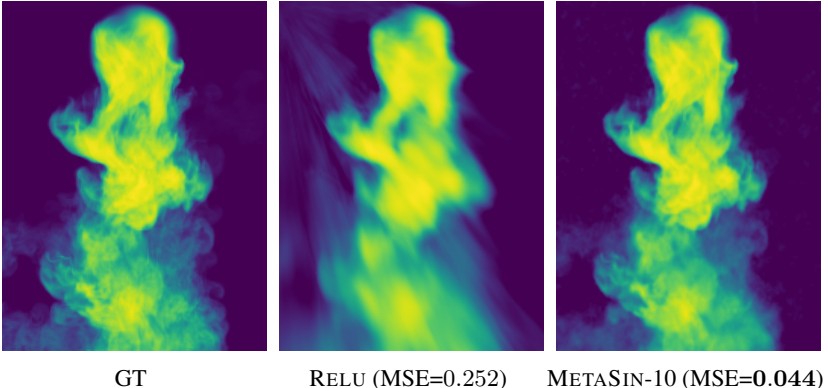

| GT | RELU (MSE=0.252) | METASIN-10 (MSE=**0.044**) |

Figure 7: False-color visualization of the reconstructed density grid where METASIN is configured with $K = 10$

## A.3 Density Overfitting using METASIN MLP

In this section we present an overfitting experiment on a 3D voxel grid. The smoke scene we used in this experiment contains $175 \times 232 \times 175$ density values within range $[0, 1]$. Following the MLP architecture used in [39], we map the 3D coordinates to the corresponding density value using RELU and METASIN activated networks. Comparisons between the METASIN activated network and the baseline RELU model are presented in Figure 7.

## B   Details of Overfitting Experiment with Convolutional Network

Table 9 shows further architectural details of the overfitting experiment we discuss in Section 3 of the main paper. We perform experiments with two base convolutional architectures, a shallow and a deep network. M-SIREN refers to the amplitude modulated SIREN [27], which has significantly more parameters due to the introduction of an additional amplitude modulation network.

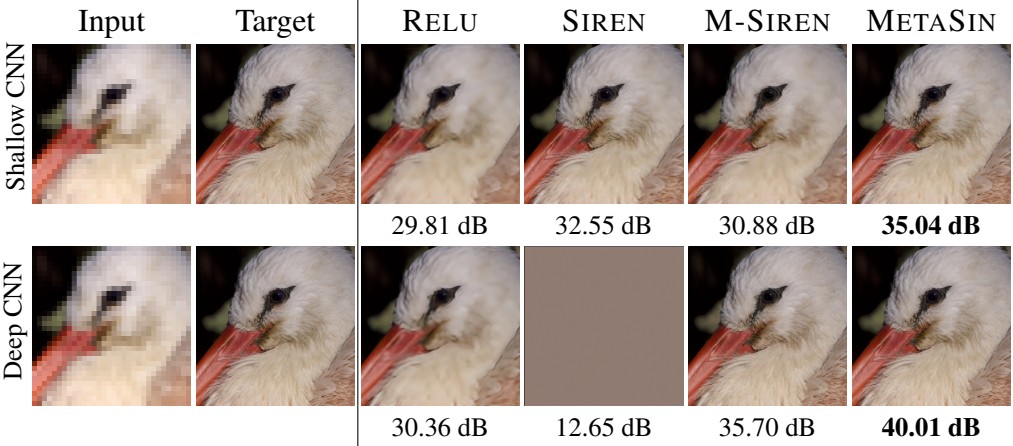

Figure 8: Peak Signal-to-Noise Ratio (PSNR) comparison of different activations employed on CNNs for image up-sampling (with 16x of upsampling factor).

Figure 8 shows a visual comparison of reconstructed images by the different versions of both base networks. Both the shallow and deep versions of the METASIN network can be trained reliably (unlike their sin activated counterparts that diverge in the case of the deep architecture). The METASIN networks come with relatively little parameter overhead (the relative overhead becomes even smaller with larger networks that have more channels, as discussed in Appendix E), and produce visually sharper results that also are closer to the reference in terms of PSNR.

| CNN Size | Activation | # Param. | Depth | PSNR (dB) ↑ |
|---|---|---|---|---|
| Shallow | RELU | 75 K (1x) | 4 | 29.81 |
| | SIREN [39] | 76 K (1x) | 4 | 32.55 |
| | M-SIREN [27] | 281 K (3.7x) | 4 | 30.88 |
| | METASIN (ours) | 82 K (1.1x) | 4 | **35.04** |
| Deep | RELU | 408 K (1x) | 13 | 30.36 |
| | SIREN [39] | 408 K (1x) | 13 | 12.65 |
| | M-SIREN [27] | 651 K (1.6x) | 13 | 35.70 |
| | METASIN (ours) | 432 K (1.1x) | 13 | **40.01** |

Table 9: Comparison of different activations employed on CNNs trained for image up-sampling (16x).

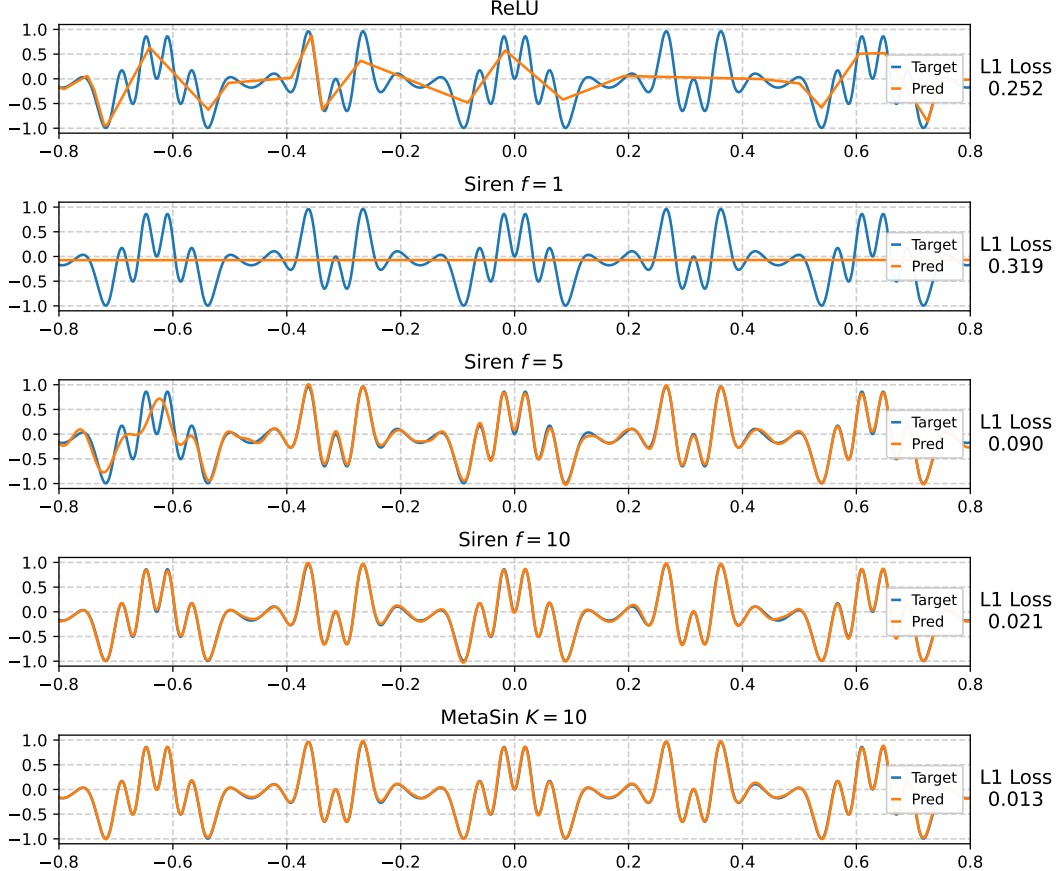

Figure 9: Representing $y(t) = \sin(10t + 5\sin(10t)) \times \sin(10\sin(10t))$ with 3 layer FC networks

## C  Sensitivity to Frequency Initialization

Although the sin activation function has promising properties as outlined in the main paper, it is known that they can be extremely difficult to train [39, 49].

We illustrate this in a toy experiment where we try to reconstruct the signal $y(t) = \sin(10t + 5\sin(10t)) \times \sin(10\sin(10t))$ when the input is $x(t) = t$. Most of the energy of $y(t)$ is distributed between frequencies between 10 and 170. We sample $t$ from the normal distribution and train fully connected networks with 1 input neuron, 3 layers having 20 neurons each, and 1 output neuron. In one of the networks, we use RELU activation in between fully connected layers except for the last

one. In the remaining two networks, we use $\sin(fx)$. In one we initialize $f = 1$, in the other one we initialize $f = 5$, and $f = 10$. We use $L1$ loss and Adam as the optimizer. The results are shown in Figure 9. As it is seen, RELU-network can reconstruct part of the signal. The sin-network with the initialization $f = 1$ does not learn at all, while the network initialized with $f = 5$ is able to match $y(t)$, and with $f = 10$ the model fits the original signal reasonably well. This suggests the importance of initialization and the difficulty of training of the sin activation function.

On the other hand, a METASIN network with the frequency parameters from Equation 4 initialized as $f_j^{[l]} = j$ and $K = 10$ effectively covers a wide range of frequencies including the one that is important to represent the signal. Consequently, using a METASIN we are able to lower the loss to $0.013$ and eliminate the trial and error process for initializing frequencies.

In Figure 10 we additionally plot the results of an experiment where we sweep through a wide range of $w_0$ values for a fully-connected SIREN network that was trained to overfit to a single image. The plot illustrates that only a narrow range among possible initial $w_0$ values results in a plausible training error, confirming the importance of initialization in SIREN networks.

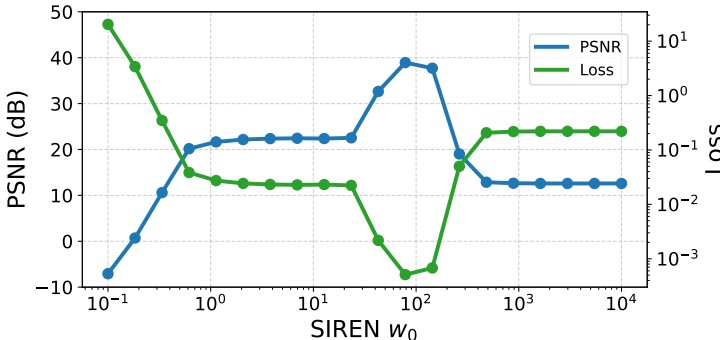

Figure 10: Sensitivity of SIREN w.r.t. base frequency $w_0$. In this figure, we reported the PSNR and MSE Loss values across a wide range of SIREN's base frequency $w_0$ values. For each $w_0$, we trained a SIREN network and showed that only a restricted set of frequencies (around $10^2$) converges to sufficiently good local minima. Specifically, in this experiment, we train the networks to up-sample the image reported in Figure 8, with an up-scale of $\times 16$ for $100,000$ iterations using the Adam optimizer with a learning rate of $10^{-5}$.

# D  Connection to Fourier Series

Well-behaved functions, such as continuous functions can be expressed as a Fourier series with finitely many terms. Let us define a function $\phi$ as a Fourier series:

$$\phi(\mathbf{x}) = a_0 + \sum_{j=1}^{K} \left( a_j \cos\left(\frac{2\pi j}{P}\mathbf{x}\right) + b_j \sin\left(\frac{2\pi j}{P}\mathbf{x}\right) \right), \tag{5}$$

where $K \in \mathbb{N}^+$ is finite, and $P$ denotes length of one period. Using the harmonic addition rule, i.e.

$$a \cos\alpha + b \sin\alpha = c \cos(\alpha + r), \text{ where}$$
$$c = \text{sgn}(a)\sqrt{a^2 + b^2} \tag{6}$$
$$r = \arctan\left(-\frac{b}{a}\right),$$

we can rewrite Equation 5 as:

$$\phi(\mathbf{x}) = a_0 + \sum_{j=1}^{K} c_n \cos\left(\frac{2\pi j}{P}\mathbf{x} + r\right). \tag{7}$$

Substituting in the identity $\cos\alpha = \sin\left(\alpha + \frac{\pi}{2}\right)$ we get:

$$\phi(\mathbf{x}) = a_0 + \sum_{j=1}^{K} c_n \sin\left(\frac{2\pi j}{P}\mathbf{x} + \frac{\pi + 2r}{2}\right). \tag{8}$$

Dropping the constant term we obtain the following:

$$\phi(\mathbf{x}) = \sum_{j=1}^{K} c_j \sin\left(f_j\mathbf{x} + p_j\right), \text{ where}$$
$$f_j = \frac{2\pi j}{P}, \text{ and} \tag{9}$$
$$p_j = \arctan\left(-\frac{b_j}{a_j}\right) + \frac{\pi}{2}.$$

From Equation 9 we can see that any Fourier Series can be expressed as a METASIN activation

$$\phi(\mathbf{x}) = c_0\text{RELU}(\mathbf{x}) + \sum_{j=1}^{K} c_j \sin\left(f_j\mathbf{x} + p_j\right), \tag{10}$$

by setting $c_0 = 0$ and the rest of its parameters $\{c_j, f_j, p_j; j \in [1, K]\}$ accordingly.

# E  METASIN Overhead

METASIN parameter overhead formula for a convolutional layer compared to RELU baseline is as follows:

$$\text{overhead} = \frac{3 \times K + 1}{\text{kernel-size}^2 \times \text{input-channels}}. \tag{11}$$

For example if we have a layer with kernel-size $= 3$ with $128$ input-channels, then overhead for $K = 10$ is $0.026$. Which means we have approximately $2.6\%$ more parameters. If input-channels is $32$ the overhead becomes $10.08\%$. On the other hand for kernel-size $= 5$ and $512$ input-channels the overhead is $0.24\%$.

# F  Details on C++/CUDA Implementation

To account for the inefficiency of a native Python API implementation of METASIN we implemented custom-optimized fused CUDA kernels for both forward and backward functions of the MetaSin activation in C++. Our implementation can be integrated into the PyTorch and TensorFlow Python APIs. Throughout the development process we also tested native automatic compilation functionalities provided by both frameworks (specifically: jit, torch.compile for PyTorch, and jit and XLA in TensorFlow). While notable improvements over the baseline Python API implementation can be made through the use of these out-of-the-box facilities, we obtained the best performances in terms of speed and memory consumption using our custom-designed CUDA functions.

Some of the techniques we utilized in our code are as follows: in order to optimize the memory footprint and the inference speed of the METASIN, we remove the intermediate quantities that the autograd engine computes and instead compute the output and gradient tensors directly from the input and the METASIN parameters. Moreover, we further optimized the computation speed with improved caching and reduction strategy on warp and block levels. Meanwhile, we have exploited a 2-level reduction strategy based on *pairwise summation* in our backward kernel. In this way, we could avoid numerical errors in gradient tensors and achieve comparable accuracy to autograd in float32 precision.

The reduction in computational overhead using the aforementioned optimizations enabled running the compute-intensive experiments we present throughout the paper in a feasible manner, and we believe demonstrates the viability of METASIN activations for most practical tasks. That being said, other optimizations that we were not able to explore due to time constraints could help reduce the overhead even further, which is an interesting direction for future research.

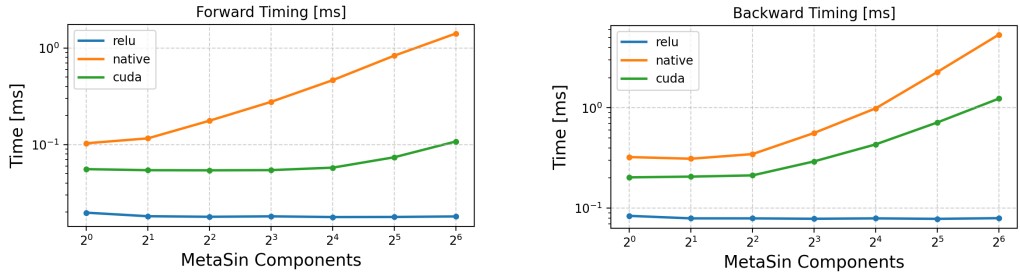

Figure 11: Effect of METASIN components to latency.

We also investigated the computational scaling behavior of our implementation with respect to the number of $sin$ components $K$. To this end, we ran a benchmark experiment, in which we compared forward and backward latencies of RELU, METASIN Native, and METASIN CUDA functions on the same input tensors using different $K$ values. The results can be found in Figure 11.

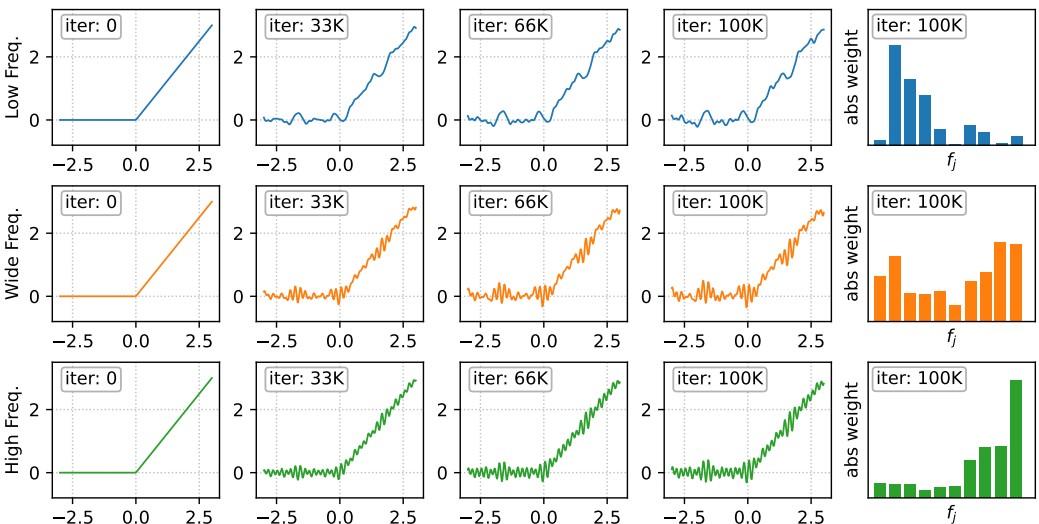

Figure 12: METASIN activation shape evolution over time and final distribution of frequency parameters. In this figure, we report the behavior of three activated neurons (of the same layer) over the training process.

## G Evolution of METASIN Activations during Training

Figure 12 shows illustrative examples of the evolution of METASIN activations during the course of training in a toy example. In this experiment the weight of the RELU component is set to $1$, whereas the amplitudes of all $sin$ components are set to $0$. Thus a METASIN activation starts out as RELU, but then evolves into various shapes depending on the training of the shape parameters via backpropagation. It is worth noting that the shape of a METASIN activation tends to not change after the initial phase of training, which agrees with the observation that weights change very slowly in overparameterized networks. Moreover, the different rows in the figure show three different METASIN neurons of the same layer that interestingly converge to a complementary frequency response that can capture different frequency components of the input signal.

To shed some light on the shapes that METASIN activations take in a more practically relevant setting with and without KD-Bootstrapping, we additionally produced a visualization of the METASIN shapes from the resampling network described in Section 4.1. The METASIN shapes are obtained from the

first and last blocks of models trained from scratch and with KD Bootstrapping. This visualization in Figure 13 shows that, while there is significant local variation between individual METASIN activations, globally the rough RELU shape is still discernible even without any involvement from the RELU teacher.

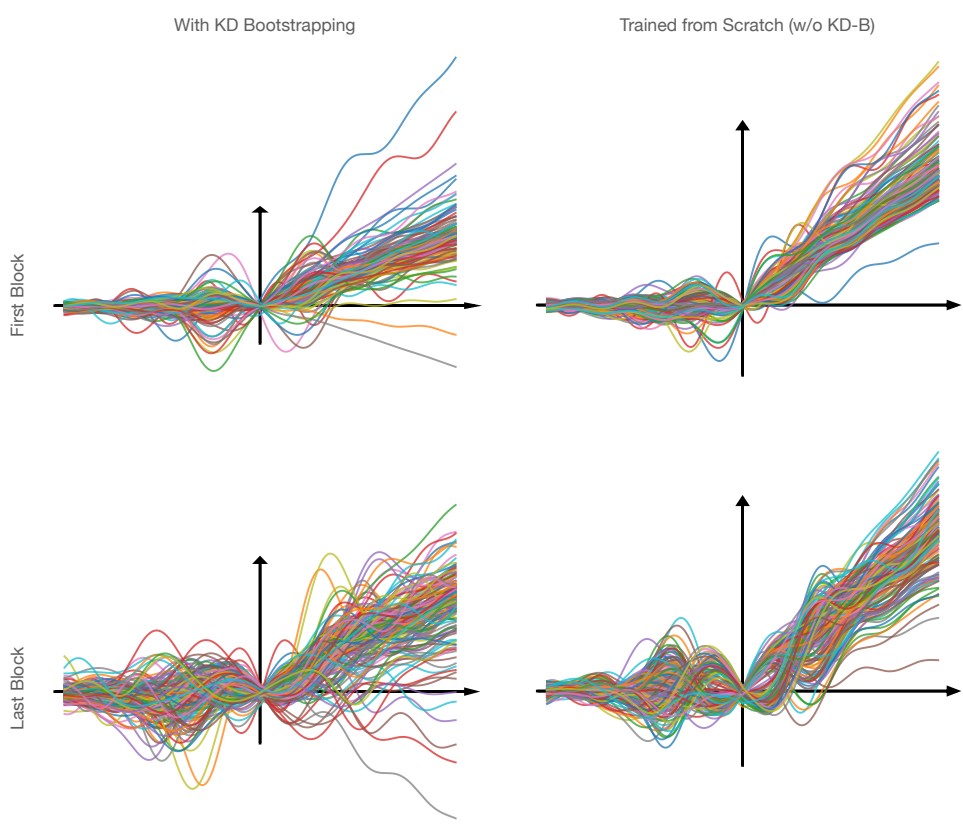

Figure 13: Visualization of METASIN shapes with and without KD bootstrapping.

## H   Comparison with Ensemble Activations

We ran additional experiments using ensemble activations as described in a recent survey [3], namely Adaptive Blending Units (abu), Variable AFs (vaf), Adaptive AFs (aaf) in the same experimental setting that we used to generate the first row of Figure 1 in the main paper. We present the best results we obtained from multiple runs with different initializations in Table 10, also including RELU and METASIN as references:

|          | abu   | vaf   | aaf   | RELU  | METASIN |
|----------|-------|-------|-------|-------|---------|
| PSNR[db] | 29.79 | 28.98 | 29.06 | 29.81 | **35.04** |

Table 10: Comparison of various enseble activations in an image overfitting setting.

We also ran two experiments in image resampling setting using abu and aaf versions of the model from Section 4.1, which we present in Table 11:

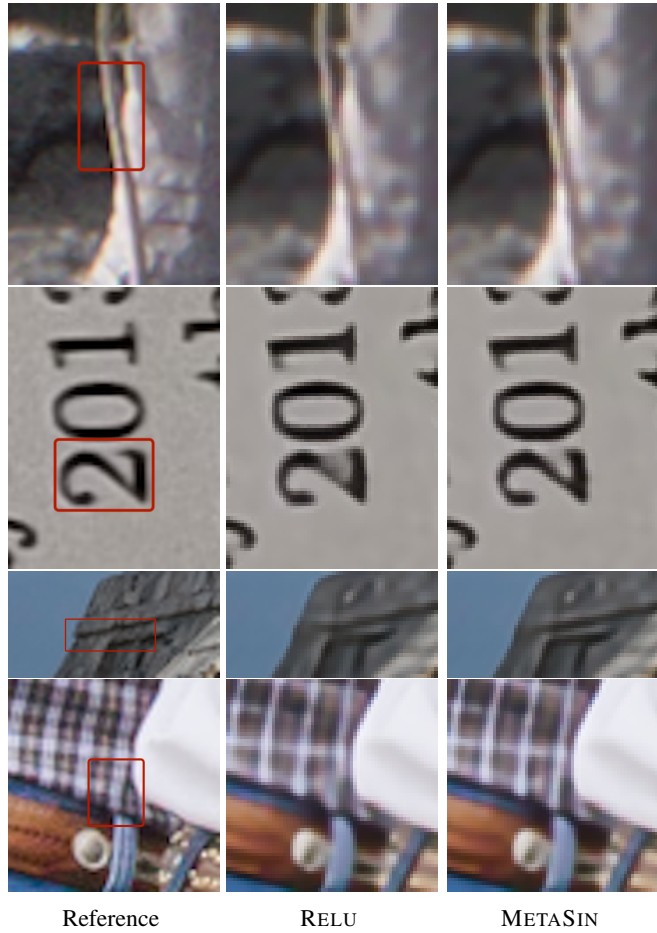

| Reference | RELU | METASIN |
|:---:|:---:|:---:|

Figure 14: Additional upsampling results comparing the state-of-the-art model with RELU activations [5] and the corresponding METASIN network.

|  | Activation | | | |
|---|---|---|---|---|
| Factor | abu | aaf | RELU | METASIN w/ KD-B |
| $P_{\times 2}$ | 33.02 | 32.97 | 33.03 | **33.26** |
| $P_{\times 3}$ | 29.31 | 29.28 | 29.36 | **29.58** |
| $P_{\times 4}$ | 27.09 | 27.10 | 27.09 | **27.29** |

Table 11: Comparison of various ensemble activations for image resampling

## I   Effect of Setting $K$ to a high number

We note that further improvements in terms of model accuracy can be made by pushing $K$ beyond what we present in the paper. We did not go beyond 10-12 as throughout our experiments we observed that doing so often resulted in diminishing returns. As a more concrete example we provide results from a resampling experiment in Table 12 (Note that these results are from a preliminary run with slightly inferior hyperparameters compared to the corresponding experiments we report in the main paper, and the training was stopped prematurely after 900K iterations instead of the full 2M - hence the lower PSNRs compared to Table 2). These results suggest that choosing $K > 10$ would indeed make sense in cases where the absolute best quality is desired and additional latencies are tolerable, but overall $K \approx 10$ seems to be the happy medium.

| Activation/Upsample Factor | $P_{\times 2}$ | $P_{\times 3}$ | $P_{\times 4}$ |
|---|---|---|---|
| METASIN-10 | 33.09 | 29.42 | 27.17 |
| METASIN-20 | 33.14 | 29.45 | 27.20 |

Table 12: The effect of setting $K$ to 10 and 20 on resampling quality

## J  Classification Experiment with METASIN Model Trained from Scratch

To investigate the effect of KD-Bootstrapping in the classification setting, we trained various Wide-ResNets with original RELU activations and with METASIN activations entirely from scratch and without using KD-Bootstrapping. Table 13 presents test accuracy on CIFAR-100.

| Model/Activation | WRN-16-2 | WRN-28-2 | WRN-40-2 | WRN-40-4 |
|---|---|---|---|---|
| RELU | 72.85 | 74.82 | 75.95 | 78.99 |
| METASIN w/o KD-B | 71.82 | 73.88 | 75.44 | 78.44 |

Table 13: Comparison of METASIN and RELU models in image classification with training without KD-Bootstrapping. In this case METASIN models only become advantageous when train using KD-Bootstrapping, underlining the importance of the proposed training methodology.

In accordance with the above discussion these results underline the role of KD Bootstrapping for achieving the best results with METASIN networks. To give a concrete example: as we show above, when training from scratch WRN-16-2-METASIN at 71.82 accuracy is behind its RELU counterpart (WRN-16-2-ReLU) with accuracy 72.85. On the other hand, the last column of Table 6 shows that by distilling from a WRN-40-2 teacher, the accuracy of WRN-16-2-METASIN can be brought up to 74.10, whereas WRN-16-2-RELU achieves 73.65 accuracy using the same procedure.

## K  Resampler Architecture and Additional Results

Our network architecture is based on the design proposed in [5]. We utilize a ProSR feature extractor [45], consisting of 3 residual blocks with 4 dense blocks within each residual block and a growth rate of 40 and 160 channels. The subsequent prediction layer is implemented as an MLP with 4 hidden layers and 256 neurons. In our experiments, this prediction layer remains unchanged. We apply METASIN activations only to the convolutional feature extractor. We present additional results in Figure 14 for the resampler experiment presented in Section 4.1.

## L  DPCN Denoiser Architecture and Additional Results

The design of the U-Net architecture we use in Section 4.2 is detailed in Table 14. We also present

| Scale | Encoder ResBlocks | Encoder Channels | Decoder ResBlocks | Decoder Channels |
|---|---|---|---|---|
| 0 | 2 | 64 | 2 | 64 |
| 1 | 2 | 128 | 2 | 128 |
| 2 | 2 | 256 | 2 | 256 |
| 3 | 2 | 256 | 2 | 256 |
| 4 | 2 | 256 | 2 | 256 |

Table 14: The Configuration of the Standard DPCN U-Net. Scale 0 is the full-resolution scale. From one scale to the next one, the image resolution is halved in both dimensions.

additional results in Figure 15 to the results we presented in Section 4.2 in the main paper.

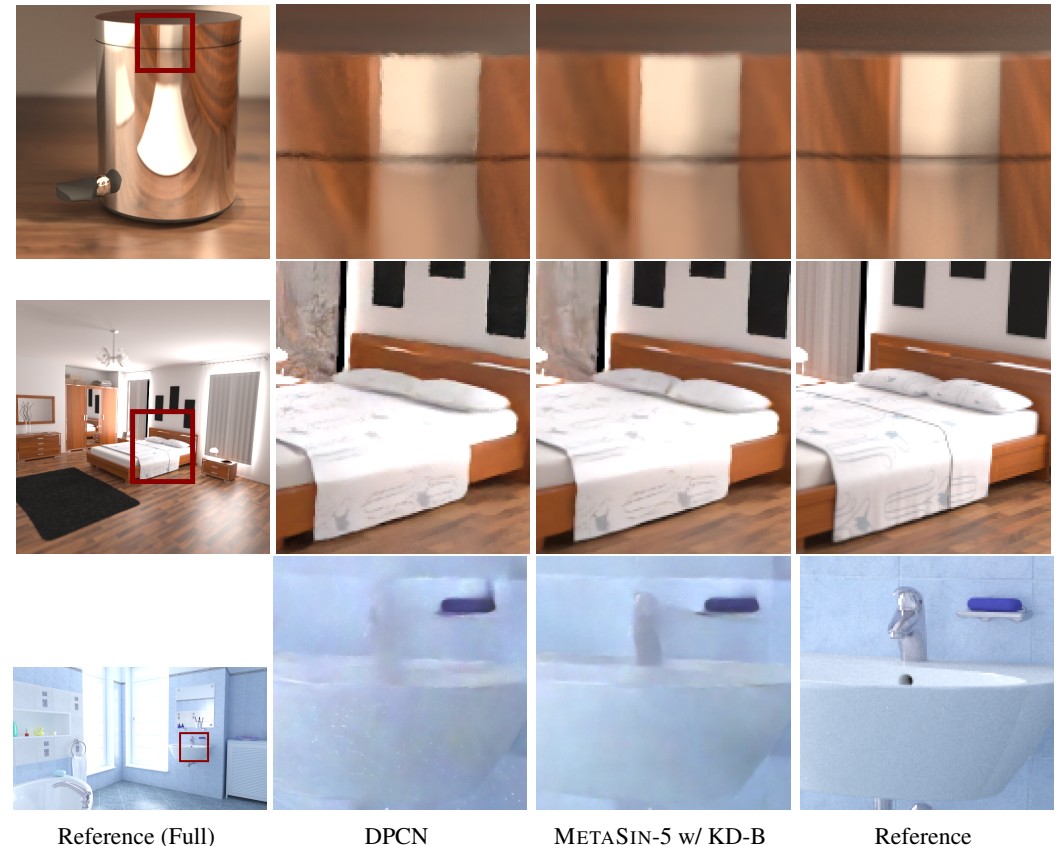

| Reference (Full) | DPCN | METASIN-5 w/ KD-B | Reference |

Figure 15: Additional results to Section 4.2

## M    KPAL Denoiser Results

In order to factor out the potential effects of kernel prediction to our denoising experiments, we utilized a direct prediction version of the state-of-the-art Monte-Carlo denoiser presented in [38]. In this section we present results of training a METASIN version of the original kernel prediction network. The improvement over the baseline RELU network remains significant in this experiment across all metrics. An interesting observation from this study was the significantly reduced effect of KD-Bootstrapping, which we eventually decided not to employ to produce the results in Table 15. We hypothesize that the lesser effect of KD-Bootstrapping in this case can be attributed to kernel prediction, which however requires further investigation.

| Metric | KPAL | METASIN-10 (Glorot) |
|--------|------|---------------------|
| SMAPE | 3.098 | **2.966 (-4.26%)** |
| flip_loss | 0.934 | **0.893 (-4.39%)** |
| 1-MS_SSIM | 2.893 | **2.622 (-9.37%)** |
| 1-SSIM | 6.227 | **5.934 (-4.71%)** |

Table 15: Comparison of the state-of-the-art kernel predicting denoiser [38] and its significantly improved METASIN version.

## N    Failure Cases

While our preliminary experiments on using METASIN activations with MLPs generally resulted in improvements over corresponding baselines, we also observed some failure cases. A particular

experiment where the tested METASIN network did not perform well was on signed distance field (SDF) overfitting. In this case, the SIREN reproduction of the ground-truth SDF is better at fine scale details than the corresponding METASIN reproduction. We hypothesize that this discrepancy in performance is due to the gradient loss that SIREN employs and the METASIN network lacks due to not being continuously differentiable. We defer further investigation of this limitation to future work.

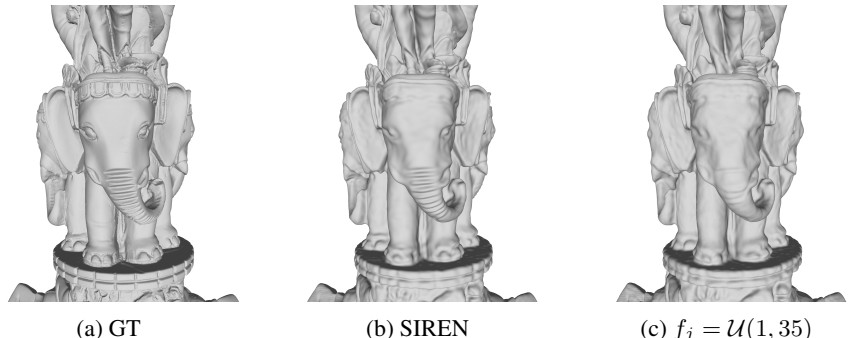

(a) GT                      (b) SIREN                      (c) $f_j = \mathcal{U}(1, 35)$

Figure 16: SDF overfitting on *Thai Statue* [1]. Here the SIREN network produces a better representation of the underlying 3D shape than a comparable METASIN activated MLP.

