# OpenReview forum: "Empowering Convolutional Neural Nets with MetaSin Activation"
_NeurIPS.cc/2023/Conference — NeurIPS 2023 poster_

### Official Review · Reviewer_pU6A · 2023-06-23

**Soundness:** 3 good
**Presentation:** 3 good
**Contribution:** 3 good
**Rating:** 6
**Confidence:** 4

**Summary:**

The paper proposes MetaSin, a new activation function for deep learning. MetaSin essentially consist of a relu function plus a sum of parametrized sin activations functions. The MetaSin function is developed specifically to work in the domain of image prediction. The paper presents multiple experiments that support MetaSin as the new state-of-the-art activation function, that also investigates different training setups and hyperparameters.

**Strengths:**

Originality:
The work seems original and novel

Quality:
The paper is of high quality and provides both a more theoretical reasoning for why MetaSin should be able to perform better than ReLU, but also backs this up with multiple experiments.

Clarity:
The paper is clearly written and easy to follow.

Significance:
As the authors mention themself in their "Broader Impact" section, a potential new SOTA activation function can have a large impact on the community with the potential to increase performance across a large selection of tasks and models.

**Weaknesses:**

While I agree with the underlying hypothesis that ReLU networks are most commonly used and therefore it is the most relevant baseline to compare to, I still think the others baselines are relevant, especially for section 5.3 on image classification. The statement in L52-53 is simply wrong, and misrepresent the conclusion in [1]. It is correct that ReLU is sometimes better, but in [1] it is only true for machine translation. For image classification Swish seems to be the better choice, and I am really missing this as a baseline in at least section 5.3.

In relation to that in L55-56 the authors mention that most other activation functions are untested on image prediction. In that case it seem obvious that the authors could have tested out more baselines than ReLU.

In general standard deviations are missing for most experiments to show if the results are indeed significant at all. I would be therefore be very careful using the term "state-of-the-art".

Regarding section 3.2: more details on the CUDA implementation, specifically regarding the kernel fusion, and what the actual gradient of the MetaSin operator is missing

[1] Prajit Ramachandran, Barret Zoph, and Quoc V. Le. Searching for activation functions, 2017.

**Questions:**

In addition to some of the weaknesses pointed out in the previous section, I would like the authors to answer:

1. Table 1: what are the computational scaling behaviour of K? It seems important as it potentially could indicate if there is a performance-computationally trade-off for this hyperparameter.

2. Table 5: It seems like increasing K is beneficial for performance. I wonder how far this behaviour can be pushed. What about increasing it further, like 20 or even higher?

Smaller correction:
L88: there is missing an end bracket in the equation.

**Limitations:**

I think the authors are fairly on point in their limitations section that MetaSin.

---

> ### Author Rebuttal · Authors · 2023-08-09
>
> **Swish as a Baseline for Classification**
>
> As the reviewer suggested we ran the image classification experiments reported in Table 6 using Swish versions of the baselines. We report the validation accuracies, in comparison to the MetaSin and ReLU results from earlier:
>
> | Teacher  | WRN-40-2  | WRN-40-4 | WRN-28-2 | WRN-40-2 |
> | :--------:                | :-------:       | :-------:       | :-------:      | :-------:       |
> | **Student**  | **WRN-40-1**  | **WRN-40-1** | **WRN-16-2** | **WRN-16-2** |
> | ReLU                  | 73.39          | 70.53          | 71.76         | 73.65         |
> | MetaSin              | **73.74**          | **72.55**         | 72.33        | **74.10**         |
> | Swish	      | 73.28	    | **72.54** 	 | **72.98**        | 72.60       |
>
> We’d be happy to update Table 6 with the additional Swish baseline and modify L255 as “... MetaSin student networks consistently outperform their ReLU counterparts, *and in some cases also the Swish student networks*, …”. We believe that the latter part  “... motivating further investigation of using MetaSin activations in classification tasks” is still appropriate and would keep it unchanged.
>
> We also greatly appreciate the reviewer pointing our attention to the statement in Lines 52-53. In fact, Swish outperforms ReLU in the image classification experiments presented in Ramachandran et al. 2017, whereas performing comparable to ReLU in others. We will replace the text in those lines accordingly with the following:
> “... A comparative study of these activations re-affirmed ReLU as a strong baseline, while suggesting Swish might be a better alternative for image classification.”
>
>
> **Significance of Experiments**
>
> To investigate this while staying within our compute budget, we ran an experiment on a slight variation of the resampling model from Section 4.1, which uses the same loss, batch and patch sizes for training, and the exact same feature extractor as the original model. The only difference is that the model is trained for 2x upsampling (instead of generic resampling) and the MLP part following the feature extractor is different to accommodate that. As a reminder, we only use MetaSin activations to replace ReLUs in the feature extractor, both in the resampling model we presented in Section 4.1, and in the newly trained upsampling model. The mean and standard deviation computed from 5 independent training runs are:
>
> | Upsampling Model  | PSNR [db]  |
> | :--------:                | :-------:       |
> | ReLU                  | 31.82    $\pm$ 0.02   |
> | MetaSin (mean/std) | 32.00 $\pm$ 0.01  |
>
> On the denoising side, unfortunately due to the resource intensive training process of the denoiser model from 4.2, repeating experiments was not an option. That being said, in Tables 3, 4, and 5 we present a total of 9 different runs that differ in their initializations, number of *sin* components, and use of KD-B. In all these experiments MetaSin models consistently outperform the DPCN baseline presented, often by a large margin. Additionally, in Appendix I we report similar levels of improvement from a kernel prediction version of the same denoiser augmented with MetaSin. We firmly believe that these consistently strong improvements are very unlikely to be due to randomness in the training procedure.
>
> **Details on CUDA kernel**
>
> As another reviewer also asked for more details, we address this in the Author Rebuttal section
>
> **Computational scaling behavior of K**
>
> We ran a benchmark experiment, in which we compare forward and backward latencies of ReLU, MetaSin Native, and MetaSin CUDA functions on the same input tensors using different K values. The results can be found in the pdf file attached to the Author Rebuttal section. As the Reviewer suggested, the results show that the computational cost gets higher as we increase K.
>
>
>
> **Effect of setting K to a higher number**
>
> The reviewer is correct that further improvements in terms of model accuracy can certainly be made by pushing K beyond what we present in the paper. We did not go beyond 10-12 as  throughout our experiments we observed that doing so often resulted in diminishing returns. As a more concrete example we provide results from a resampling experiment below:
>
> | Activation / Upsample Factor | x2  | x3 | x4 |
> | :--------:      | :-------:       | :-------:       | :-------:  |
> | MetaSin-10 | 33.09       | 29.42      | 27.17     |
> | MetaSin-20 | 33.14       | 29.45      | 27.20     |
>
> (Note that this is from an older, but in our opinion still sufficiently representative run with slightly inferior hyperparameters, and the training was stopped prematurely after 900K iterations instead of the full 2M - hence the lower PSNRs compared to Table 2)
>
> These results suggest that choosing K > 10 would indeed make sense in cases where the absolute best quality is desired and additional latencies are tolerable, but overall K $\approx$ 10 seems to be the happy medium.

---

> > ### Comment · Reviewer_pU6A · 2023-08-15
> > **Acknowledgement**
> >
> > Thank you very much for your detailed response to my concerns.
> >
> > I greatly appreciate the additional clarifications and the additional experimental results provided (even if some of the are a bit out-of-date), which answers all of the questions I have regarding the paper. Based on concerns raised by other reviews and the strong answers you have given them I will raise my score by one.
> >
> > I would still greatly advice that some of the details regarding the custom cuda implementation (and thoughts about it) should be included in a appendix for the final version of the paper.

---

> > > ### Author Response · Authors · 2023-08-15
> > >
> > > We are happy that we were able to address all of the Reviewer's questions in our rebuttal. As suggested, we will add a discussion on the CUDA implementation in the final version of the paper.

---

### Official Review · Reviewer_C7v1 · 2023-06-29

**Soundness:** 3 good
**Presentation:** 4 excellent
**Contribution:** 3 good
**Rating:** 6
**Confidence:** 4

**Summary:**

The paper proposes a modification of the sine activation function and show that this results in improved performance, compared to RELU and others. The new activation, called METASIN, is a superposition of RELU with several sinusoidal functions, and is motivated by the observation that RELU has a spectral bias to low frequencies. To ensure that training is stable in deep networks, the authors propose a distillation approach, where the activation function parameters, such as amplitudes, are initialized based on a teacher RELU model. The authors show that by combining these two techniques, they outperform RELU and achieve state-of-the-art results on denoising and image resampling tasks.

**Strengths:**

- The paper introduces a new activation function that can be of interest to the community, which contains RELU as a special case.

- The authors provide an optimized implementation of the activation function that is 3 times faster compared to native PyTorch implementation without increasing the memory overhead. In addition, the authors report that the new activation increases overall training compute by only 3% compared to RELU. However, this does not account for the impact of distillation, which seems to be needed.

- The empirical results are strong, achieving SoTA on a few tasks, such as image resampling and denoising.

- The paper is well-written and is easy to follow.



**Weaknesses:**

- It seems that all experiments are carried out with distillation. The authors state that they also apply distillation to RELU networks but how does that work when RELU does not contain any shape parameters? If the authors introduce an amplitude to RELU, its effect may vanish when also using normalization layers, which may explain why the authors do not observe any impact of distillation in RELU networks. This needs to be clarified.

- The experimental setup seems to be different in each experiment. For example, the authors use KD-bootsrapping during the first 5% of training in the denoising experiment but switch to 10% in image resampling. They also use a different initialization of the frequencies. Are these chosen based on a separate validation set? It is not clear if this is the case.

- The authors claim in the abstract that the improvement is obtained by "simply replacing the activations". However, the new activations contain trainable parameters that are also initialized/trained using distillation. It is not a simple matter of replacing the activations.



**Questions:**

- In Line 160, the authors say that the KD Bootstrapping approach comprises only 10% of the total training budget. However, you are also training another model with RELU activations. Wouldn't this increase the training compute by about a factor of 2? Or is the 10% here only for training METASIN shape parameters based on the teacher model?

- How do you select the KD-bootsrapping duration in each experiment? Is it based on a separate validation split? If so, please mention this explicitly in the paper.

- References are missing parentheses throughout the paper. Consider replacing \cite with \citep.

---

> ### Author Rebuttal · Authors · 2023-08-09
>
> **Distillation and ReLU Networks**
>
> The reviewer is correct that our best results in both image resampling and denoising applications are obtained using distillation, specifically KD Bootstrapping as discussed in Section 3.1 of our manuscript. In short: the role of distillation in these experiments is to stabilize training in the early phases and guide the network towards good local minima. Later during the training we turn it off and allow the network to learn the final solution independently without any additional constraints. This is a specific use case for distillation tailored to training MetaSin activations.
>
> In general though, Knowledge Distillation is used to enhance accuracy of a student network (usually with fewer parameters) by utilizing a pre-trained teacher network (usually with more parameters). The image classification experiment in Section 5.3 is carried out in this general knowledge distillation setting, where the teacher-student pairs and corresponding accuracies are listed in Table 6. Thus, to make the comparison between MetaSin and ReLU student networks fair, we apply distillation to both student variants. This experiment shows that MetaSin student networks are capable of imbibing more information than their ReLU counterparts from identical teacher models.
>
> We’d be happy to modify the text accordingly to clarify the use of knowledge distillation in the aforementioned experiments.
>
> **Frequency initialization**
>
> Through grid search across many experiments we performed for this work (on fixed validation sets), we empirically found that initializing the frequency shape parameters of MetaSin activations as $f_j = j$, where $j \in [1, K]$ ($K$ being the number of *sin* components) works well in most of our experiments involving convolutional models, which is what we recommend in Section 3.1 as a sensible default. That being said, as with any hyperparameter, slight improvements can be made by tweaking the default value: For instance the last two columns of Table 3 show that $f_j = j/2$ yields better results when training the DPCN denoiser described in Section 4.2.
>
> **Duration of KD Bootstrapping**
>
> In order to produce our denoising and resampling results we performed KD Bootstrapping for 200K iterations, which correspond to 5 and 10% of the total training time, respectively. Similarly to the frequency initialization, we determined that the 5-10% of total training iterations we recommend as default in Section 3.1 through our observations across various experiments we performed during this work. This duration has been sufficient to stabilize training and guide the network towards good local minima in the beginning, and we observed from that point onwards more KD Bootstrapping steps do not improve the results.
>
> **Total training time with KD Bootstrapping**
>
> The percentage in Line 160 refers to the ratio of the total training iterations in which we utilize distillation, i.e. incur an additional distillation loss during training. As the computational overhead of the distillation loss is minimal the total training time with or without KD Bootstrapping is roughly the same. That being said, in both our denoising and resampling experiments we had access to the trained baseline ReLU networks. If that is not the case, a ReLU network needs to be trained beforehand, and this should be included in the training budget. We would be happy to modify the text to clarify this point.
>
> **”Simply replacing the activations”**
>
> We address this more in detail in the Author Rebuttal section as another reviewer raised a similar concern. In short: Although the procedure for replacing the activations (including re-training) is described later in the text in Lines 140-143, we fully agree that our phrasing in Line 12 might be misleading and will modify the text accordingly.

---

> > ### Comment · Reviewer_C7v1 · 2023-08-13
> > **Acknowledgement**
> >
> > Thank you for answering the questions.
> >
> > I do believe that the paper should clarify the role of distillation in the paper, and avoid phrasings like in Line 12 and the statement about the impact on compute, both of which may give a false impression. I'm happy that you agree to revise those. For the duration of KD Bootstrapping, I'm satisfied with the answer, but it's perhaps worth clarifying in the paper as well.

---

> > > ### Author Response · Authors · 2023-08-14
> > >
> > > We are happy that our rebuttal answered the Reviewer’s questions, and the revisions we proposed for the text were found suitable. In addition to the revisions already discussed, we will extend the text in Line 159-160 with the points we make in our answer to “Duration of KD Bootstrapping”, as the Reviewer suggested.

---

### Official Review · Reviewer_9soJ · 2023-06-30

**Soundness:** 3 good
**Presentation:** 3 good
**Contribution:** 2 fair
**Rating:** 6
**Confidence:** 5

**Summary:**

**SUMMARY AFTER REBUTTAL**: the authors have addressed most of my concerns and I have increased my score during the rebuttal phase. I believe the novelty of the paper is small if moving beyond their specialized subfield, which is why the overall score remains low.

---

The paper proposes a variant of the sin activation function proposed in SIREN, specifically for image reconstruction and image denoising.

Instead of using a single sin function, they consider a linear combination of a ReLU function and multiple sin functions, where the weights of the linear combination and the parameters of the sin functions are shared across layers and trained with back-propagation. The networks are trained via a simple knowledge distillation procedure after a drop-in replacement of the activation functions of the corresponding ReLU networks.

A series of experiments shows better image reconstruction capabilities with a very small overhead (thanks to a custom CUDA kernel) on two major benchmarks.

**Strengths:**

- The paper is very well written and easy to follow.
- Experiments are good (see some remarks below), and the improvements are consistent.
- The motivation for the proposed AFs is only discursive, but clear.
- As far as I know, this formulation is novel with some caveats (see below).

**Weaknesses:**

- My main concern is that the idea of building an AF from a linear combination of base AFs whose weights are trained is quite known in the literature. Considering for example the survey on AFs by Apicella et al., 2021 [A survey on modern trainable activation functions], they have an entire chapter on this idea including Adaptive AFs, Variable AFs, Kernel-based AFs, Adaptive Blending Unit, Adaptive Piecewise Linear Unit, etc.
- The core contribution of this paper is to apply this idea to sin AFs, i.e., considering a different base set from the papers cited above (which combined other types of functions, such as kernels with different weights, ReLUs and sigmoids, etc.), showing this is useful in the context of image reconstruction. While interesting, this is a niche result that is best suited for a smaller conference or journal with applicative interest.
- In addition, the authors are only comparing to standard AFs, although they consider a "MReLU" AF which seems similar to the ensemble AFs described above. It would be helpful to have a stronger comparison to ensemble AFs.

Note that a similar paper vertical on PINNs was rejected from last ICLR (https://openreview.net/forum?id=MpGP-z07TmM) due to similar reasons.

An additional weakness is that the authors claim their method is a complete drop-in replacement, while it requires a KD step to work correctly. I believe these claims should be amended.

**Questions:**

Apart from the previous weaknesses, I would be curious to see more details on their CUDA kernel. Is this done manually, or is it just the result of PyTorch compile procedures?

I am also skeptical of their initialization range, since they set to 0 all weights multiplying the sin AFs. Wouldn't this prevent a good gradient flow? More plots of the resulting sin functions before and after training can improve the discussion here.

While the paper is well written, there are some citations that appear in a wrong way (e.g., "state-of-the-art resampler Bernasconi et al. [4]"). There is also a small typo on P3 ("it’s predictions").

**Limitations:**

All limitations are correctly described in the paper.

---

> ### Author Rebuttal · Authors · 2023-08-09
>
> **Comparison with ensemble activations**
>
> Throughout the paper we present comparisons against popular baselines Snake, Mish, Siren, as well as an ensemble activation we call MReLU that is similar to the Adaptive Piecewise Linear Units presented by Agostinelli et al., 2015 and mentioned in the survey by Apicella et al., 2021 (we will add citations accordingly). Tables 2 and 4 in our main paper show that MReLU consistently performs worse than MetaSin in both resampling and denoising applications. Following up on the reviewer’s comments we ran additional experiments using other ensemble activations Adaptive Blending Units (abu), Variable AFs (vaf), Adaptive AFs (aaf) in the same experimental setting that we used to generate the first row of Figure 1 in the main paper. We present the best results we obtained from multiple runs with different initializations below, also including ReLU and MetaSin as reference:
>
> | Activation |  abu | vaf | aaf | ReLU | MetaSin|
> | :--------:      | :-------:       | :-------:       | :-------:      | :-------:       | :-------:
> | PSNR [db] |  29.79      | 28.98      |  29.06     | 29.81       |  35.04 |
>
> We also were able to run two experiments in image resampling setting using abu and aaf versions of the model from Section 4.1, which we present below:
>
> | Activation | x2  | x3 | x4 |
> | :--------:      | :-------:       | :-------:       | :-------:      |
> | abu |  33.02       | 29.31      | 27.09 |
> | aaf  |  32.97      | 29.28       |  27.10 |
> | ReLU | 33.03   | 29.36   | 27.09 |
> | Metasin KD-B |  33.26 | 29.58 | 27.29 |
>
> Overall the ensemble activations we tested tend to perform roughly similar to ReLU. We would be happy to include the above data points in the final manuscript.
>
> **Rejected ICLR submission on PINNs**
>
> The main criticism that led to this decision (despite otherwise favorable reviews) appears to be the argument strongly put forth by Reviewer `E27E` stating that:
> “The physics-informed activation function (PIAC, Eq. (10)) …”, which the submission claims as a novel contribution,  “... is **identical** to, among others, the soft-normalized version of the adaptive blending unit (ABU) [1]” (emphasis by `E27E`)
> As such, we don’t believe the analogy to our work applies, since to the best of our knowledge the MetaSin formulation is novel and all of our reviewers seem to agree with us on this point.
>
> [1] Sütfeld et al., Adaptive Blending Units: Trainable Activation Functions for Deep Neural Networks, 2017
>
>
> **Results are Niche**
>
> Our main contributions in this paper include the specific formulation of the MetaSin activations that we arrive at by gradually addressing issues associated with *sin* activations as described in Section 3 in or paper, as well as the training methodology that we call KD Bootstrapping that is described in Section 3.1, which to our knowledge has not been explored in the context neither *sin*-based nor ensemble activations.
>
> While, on one hand, we focused our efforts on thoroughly investigating the use of MetaSin activations in two arguably core topics within our target domain of image prediction applications, namely resampling and denoising, on the other hand we believe that it is not unreasonable to assume other applications within this vast domain might benefit from the techniques we present in this paper. In support of this argument, we present various preliminary results throughout the paper, which show the effectiveness of MetaSin activations in NeRF models (Table 8 in Appendix) for predicting novel views, as well as in 2D (Table 7 in Appendix) and 3D (Figure 7 in Appendix) signal representation tasks.
>
> The domain of image prediction applications spans a big chunk of classical research problems in image/video processing and computer graphics/vision: segmentation, matting deblurring, tone mapping, depth estimation, view interpolation, inpainting, to name a few. Moreover, one can speculate that our work might find applications in contemporary generative models relying on diffusion models considering our strong results in denoising. Taking all these exciting directions into account, we firmly believe that our work would be interesting to the broader research community.
>
> **”Drop-in Replacement” and Details on CUDA kernel**
>
> We address both topics in detail in the Author Rebuttal section as other reviewers made similar remarks.
>
> **Initialization of MetaSin and Gradient propagation**
>
> We initialize the weights to be 0 intentionally, as we aim to prevent the activation from having arbitrary frequencies during the initialization phase. Instead, we allow the network to determine which frequencies to use during training, as we illustrate in Figure 11 in the Appendix. This initialization approach constrains the frequencies to remain relatively stable in the initial stages, akin to a Fourier series. However, as training progresses, all parameters become freely updatable, allowing the frequencies to adapt and evolve over time.
>
> **On Presentation**
>
> We noticed that the reviewer mentioned that our paper is “very well written and easy to follow” despite assigning a “fair” presentation score. Please let us know In case there are any outstanding points we can improve on the writing side.

---

> > ### Comment · Reviewer_9soJ · 2023-08-16
> >
> > I thank the reviewers for the detailed answer. To elaborate on:
> >
> > 1. "clarity score": the score takes into consideration "relation to prior work", which was missing multiple papers on combining activation functions. In addition, the description of the CUDA kernel had no details. Based on the rebuttal, I have increased the score.
> >
> > 2. "novelty": arguing on novelty is always tricky, however, from my point of view building combinations of AFs is not novel. The difference of this work with respect to the other works I cited is that, instead of combining standard AFs (eg, ReLU, tanh, ...) they combine a ReLU and sines with trainable frequencies. This is justified by their applicative domain and their analysis, but it is "niche" in the sense that it is a small modification that only makes sense in this specific set of experiments.
> >
> > I have increased my evaluation to "weak accept", as I believe the paper is clear but the contribution's strength can be discussed (as per point 2 above).

---

> > > ### Author Response · Authors · 2023-08-17
> > >
> > > We are glad that our rebuttal addressed the Reviewer’s concerns on clarity. We will revise the text accordingly to include a discussion on previous methods summarized in Apicella et al., 2021, as well as the details on the CUDA kernel.

---

### Official Review · Reviewer_N8tS · 2023-07-05

**Soundness:** 3 good
**Presentation:** 3 good
**Contribution:** 2 fair
**Rating:** 6
**Confidence:** 4

**Summary:**

The authors of this paper propose a new activation function, which relies on a parametrized sinusoidal function instead of only a piecewise linear function. The authors show how this function can lead to performance improvements in the setting of denoising.

**Strengths:**

- The authors propose a novel formulation for an activation function, relying on a sum of sines of varying amplitude, frequency and phase (all of which are parameters learned during training). The formulation of this activation function is novel, as far as I am aware.

- I appreciate the explanation that the authors provide on the usefulness of the sine activation, namely how it enables the network to capture higher frequency components. Moreover, I appreciate the inclusion of the link to Fourier series in the Appendix, and I think it may be an interesting topic that the authors may want to elaborate on.

- The authors have performed several experiments to evaluate the performance of their method, not only in their main problems of interest (denoising and resampling) but also more usual classification tasks. They have also made the effort to write optimized code for their proposed activation function (although I highly encourage releasing this to assure reproducibility).

**Weaknesses:**

The main weaknesses of the paper are the following (mostly concerning the experimental setup):

- While the experiments cover a wide range of tasks, they contain a weakness in that most of them rely on starting from a pretrained network to achieve the best results. While the authors acknowledge this limitation, it would nevertheless improve the paper if results for training without the initial model were included.

- Related to the above, the classification experiment only considers the setting where the MetaSin network learns from a pretrained teacher. I believe it would be interesting to include a classification experiment where the new architecture is trained from scratch.

- In the Appendix, the authors include a plot that shows how the weights of the activation change during the course of the training. We can see that most of the weight is in the ReLU activation (in other words, the initial form of the MetaSin activation). While this is not inherently bad, it makes the benefit induced by the new activation less clear in my opinion. I believe that this may also stem from the initialization via a pretrained model (which has already converged while using a ReLU activation), and thus may be alleviated if trained from scratch. I would greatly appreciate it if the authors were able to examine this.

**Post rebuttal comment**: As mentioned in my response to the author rebuttal below, while most of my concerns have been addressed, the fact that KD bootstrapping is required remains a limitation. However, I have a generally positive view of this work, given that as the authors mentioned in their comments the use of KD bootstrapping for stability, while limiting, is still an interesting approach.

**Questions:**

I would be grateful if the authors could elaborate on the issues encountered during training from scratch with their proposed activation. The way the paper is written, it seems to me that training with MetaSin would suffer from the same problems as previous work using sinusoidal activations e.g. SIREN, which is somewhat surprising given that MetaSin contains ReLU as a special case.

**Limitations:**

The authors have adequately addressed the limitations of their work. Furthermore, I see no immediate negative societal impact arising from this work.

---

> ### Author Rebuttal · Authors · 2023-08-09
>
> **MetaSin without KD Bootstrapping from a pre-trained network**
>
> During our experiments we had a chance to confirm firsthand the well-known difficulties associated with training *sin*-based activations, especially when utilized in convolutional networks. These difficulties may even lead to divergence in training: an example can be found in Figure 1 row 2. We haven’t encountered divergence issues when training MetaSin networks with or without KD bootstrapping. That said, our experiments in which we did not utilize KD bootstrapping often yielded sub-par results. For instance, in Table 2 see the last two columns: Removing KD Bootstrapping leads to worse results on average in resampling. The same behavior can be observed likewise in Table 4 for denoising. We hope that these two experiments, where we compare our full method (that is, MetaSin with KD-B), with an ablation where we do not utilize a pre-trained network (MetaSin w/o KD-B) will help the readers to put the contribution of KD Bootstrapping into proper context. If needed, we would also be happy to modify the text to point the reader’s attention to these experiments.
>
> **Classification experiment with MetaSin from scratch**
>
> To investigate this we trained various Wide ResNets with original ReLU activations and with MetaSin activations entirely from scratch and without using any Knowledge Distillation. The table below presents test accuracy on CIFAR-100:
>
>
> | Model/Activation  | WRN-16-2  | WRN-28-2 | WRN-40-2 | WRN-40-4 |
> | :--------:                | :-------:       | :-------:       | :-------:      | :-------:       |
> | ReLU                    | 72.85          | 74.82          | 75.95         | 78.99         |
> | MetaSin from scratch | 71.82          | 73.88         | 75.44         | 78.44         |
>
>
> In accordance with the above discussion these results underline the role of KD Bootstrapping for achieving the best results with MetaSin networks. To give a concrete example: as we show above when training from scratch WRN-16-2-MetaSin at 71.82 accuracy lacks behind its ReLU counterpart (WRN-16-2-ReLU) with accuracy 72.85. On the other hand, Table 6 last column of our main paper shows that by distilling from a WRN-40-2 teacher, the accuracy of WRN-16-2-MetaSin can be brought up to **74.10**, whereas WRN-16-2-ReLU achieves 73.65 accuracy using the same procedure. We’d be happy to modify the paper accordingly to ensure that our side investigation into image classification is as informative as possible.
>
>
> **Shape variation of MetaSin activations**
>
> To shed some light on the shapes that MetaSin activations take when training from scratch, we produced a visualization of the MetaSin shapes from the resampling network described in Section 4.1. The MetaSin shapes are obtained from the first and last blocks of models trained from scratch and with KD Bootstrapping. This visualization can be found in the pdf file attached to the Author Rebuttal section. The figure shows that, while there is significant local variation between individual MetaSins, globally the rough ReLU shape is still discernible even without any involvement from the ReLU teacher.
>
>
> **Issues when training from scratch:**
>
> One of the main challenges when training *sin*-based activations such as SIREN is their   dependence on initialization. In Appendix C we present an illustration of this behavior through a set of toy examples. Another challenge when training *sin*-based activations are inconsistent gradients due to complex shapes the activation can take, and large degeneracy in local minima caused by symmetries. These make the training of  deep networks especially unstable and may lead to divergence (See Figure 1. Deep CNN/SIREN).
>
> With MetaSin we avoid the aforementioned issues by having better coverage of plausible ranges of the shape parameters and introducing the additional ReLU component that stabilizes the training. As such, when training MetaSin models (either with and without KD Bootstrapping) we haven’t encountered dramatic inconsistencies in model accuracy due to initialization, nor encountered any further stability issues during training.
>
> The main issue with training from scratch without KD Bootstrapping is that in challenging real-world problems (such as training direct prediction models for denoising and resampling) the accuracy of the model is inferior to the alternative of training with KD Bootstrapping. On the other hand, in simpler tasks, such as the various overfitting experiments we present throughout the paper, models trained from scratch tend to perform just as well. Finally, in Appendix I we present an interesting finding that a kernel-predicting version of the denoiser from Section 4.2 (as opposed to direct prediction) also trains fine without KD Bootstrapping, which we hypothesize is due to the reduced dimensionality of the problem space, but nevertheless remains an interesting direction for further investigation.

---

> > ### Comment · Reviewer_N8tS · 2023-08-12
> > **Response to rebuttal.**
> >
> > Thank you very much for your detailed response to my concerns.
> >
> > I greatly appreciate the additional clarifications on the points I raised. While my concerns have been mostly addressed, from the resulting experiments it still seems that MetaSin requires bootstrapping in order to achieve good results, both in the original setting as well as in the image classification one. While this is fully acknowledged in the paper, it nevertheless remains a limitation of this work.
> >
> > As such, while I'm still positive towards this work, I am electing to keep my score for now. I am, however, interested in the discussion with the other reviewers as well.

---

> > > ### Author Response · Authors · 2023-08-12
> > >
> > > We are happy to hear that our clarifications were helpful in addressing most of the reviewer’s concerns. We in fact consider KD bootstrapping as a core ingredient of our method that helps alleviate the difficulties associated with training models with *sin*-based activations, which we elaborate on in detail in our paper. To the best of our knowledge the use of  knowledge distillation for improving training stability has not been explored before, and we have been positively surprised by the effectiveness of this relatively easy to implement technique.

---

### Author Rebuttal · Authors · 2023-08-09

We thank all the reviewers for their time and insightful comments. We are encouraged that the reviewers are unanimously leaning towards accepting our submission for publication. In the following we start by briefly summarizing our motivations, then address similar remarks by two reviewers about the phrasing we use to describe the usage of MetaSin activations, and finally provide more details on our optimized CUDA implementation.


**Motivation and broader implications**

Our motivation for this work came from the promising recent results obtained by using MLPs with *sin* activations in visual representation applications including image, video, and 3D shapes. In spite of these findings, however, in the general area of image prediction models, i.e. the large family of models that predict colors of image pixels as their output, ReLU activations still tend to be the default choice. From our own experience, this tendency stems from various issues associated with *sin* networks, such as the training stability and sensitivity to initialization, as well as the lack of methods for utilizing *sin* activations in *convolutional* networks, which enjoy heavy use in the domain of image prediction applications. The aim of the techniques we present in this paper is to pave the way to enable practitioners of this vast domain to reap the benefits of *sin* activations while maintaining the relative training stability of ReLU networks.

Our results suggest that there is room for significant improvements even in cutting-edge image prediction models, which can be leveraged through a targeted approach such as MetaSin with KD-Bootstrapping. We believe that our findings might be highly useful to practitioners in related application areas including and beyond the ones we touch on in this paper.


**MetaSin as a “drop-in” ReLU replacement** (`9soJ`, `C7v1`)

This phrase refers to the process of switching to MetaSin in existing code, which consists of replacing `relu` functions with `metasin(K)`. However, after reading their comments and revisiting our own initial text, we agree with the reviewers that it can be misleading as there are additional steps of KD-Bootstrapping and training the model and shape parameters. We will accordingly replace the two total occurrences of term “drop-in replacement” in lines 9 and 136 with “convenient replacement”, and remove the word “simply” in line 12.

**Details on C++/CUDA Implementation**  (`9soJ`, `pU6A`)

To account for the inefficiency of a naive Python API implementation of MetaSin we implemented custom-optimized fused CUDA kernels for both forward and backward functions of the MetaSin activation in C++. Our implementation can be integrated into the PyTorch and TensorFlow Python APIs. Throughout the development process we also tested native automatic compilation functionalities provided by both frameworks (specifically: jit, torch.compile for PyTorch, and jit and XLA in TensorFlow). While notable improvements over the baseline Python API implementation can be made through the use of these out-of-the-box facilities, we obtained best performances in terms of speed and memory consumption using our custom-designed CUDA functions.

Some of the techniques we utilized in our code are as follows: In order to optimize the memory footprint and the inference speed of the MetaSin, we remove the intermediate quantities that the autograd engine computes and instead compute the output and gradient tensors directly from the input and the MetaSin parameters. Moreover, we further optimized the computation speed with improved caching and reduction strategy on warp and block levels. Meanwhile, we have exploited a 2-level reduction strategy based on *pairwise summation* in our backward kernel. This way we could avoid numerical errors in gradient tensors and achieve comparable accuracy to autograd in float32 precision.

The reduction in computational overhead using the aforementioned optimizations enabled running the compute intensive experiments we present throughout the paper in a feasible manner, and we believe demonstrates the viability of MetaSin activations for most practical tasks. That being said, other optimizations that we were not able to explore due to time constraints could help reduce the overhead even further.

**Small Corrections**

Finally, we greatly appreciate that the reviewers took the time to point out various typos, spelling errors, minor latex issues, and so on. We noted them all and will fix them in the final manuscript. We address other remarks individually for each reviewer in the corresponding Official Comment sections.

---

> ### Author Response · Authors · 2023-08-15
>
> We thank once again all the reviewers for their insightful and constructive comments, and through those giving us the chance to make corrections, present further clarifications, as well as running additional experiments. We believe our paper will notably improve as a result of incorporating the revisions that have been discussed so far. Please let us know if there are further remarks and we’d be happy to address them to the best of our ability.

---

### Decision · Program_Chairs · 2023-09-21

**Decision:**

Accept (poster)

**Comment:**

The reviewers mentioned several positive aspects of the paper, e.g.:
- The paper proposes a novel formulation for an activation function, relying on a sum of sines of varying amplitude, frequency and phase (all of which are parameters learned during training).
- The experiments are promising.
- The paper is very well written and easy to follow.

However, they also raised multiple doubts about it, such as:
- There are some deficiencies in the experiments (e.g., settings, other baselines like MReLU).

All reviewers agreed that the paper could be accepted (6-6-6-6). It seems the paper is a solid submission, and I tend to agree with that.